# Retrograde Ret signaling controls sensory pioneer axon outgrowth

Adam Tuttle[1], Catherine M Drerup[1†], Molly Marra[1,2‡], Hillary McGraw[1§], Alex V Nechiporuk[1*]

[1]Department of Cell, Developmental, and Cancer Biology, Oregon Health & Science University, Portland, United States; [2]Neuroscience Graduate Program, Oregon Health & Science University, Portland, United States

*For correspondence:
nechipor@ohsu.edu

Present address: †Unit on Neuronal Cell Biology, National Institute of Child Health and Human Development, National Institutes of Health, Bethesda, United States; ‡Science Department, St. Mary's Academy, Portland, United States; §Division of Cell Biology and Biophysics, School of Biological Sciences, University of Missouri-Kansas City, Kansas City, United States

Competing interests: The authors declare that no competing interests exist.

**Abstract** The trafficking mechanisms and transcriptional targets downstream of long-range neurotrophic factor ligand/receptor signaling that promote axon growth are incompletely understood. Zebrafish carrying a null mutation in a neurotrophic factor receptor, Ret, displayed defects in peripheral sensory axon growth cone morphology and dynamics. Ret receptor was highly enriched in sensory pioneer neurons and Ret51 isoform was required for pioneer axon outgrowth. Loss-of-function of a cargo adaptor, Jip3, partially phenocopied Ret axonal defects, led to accumulation of activated Ret in pioneer growth cones, and reduced retrograde Ret51 transport. Jip3 and Ret51 were also retrogradely co-transported, ultimately suggesting Jip3 is a retrograde adapter of active Ret51. Finally, loss of Ret reduced transcription and growth cone localization of Myosin-X, an initiator of filopodial formation. These results show a specific role for Ret51 in pioneer axon growth, and suggest a critical role for long-range retrograde Ret signaling in regulating growth cone dynamics through downstream transcriptional changes.
DOI: https://doi.org/10.7554/eLife.46092.001

## Introduction

One major regulator of axon outgrowth and pathfinding is signaling through neurotrophic factors. Of these, the best studied neurotrophic factors are Nerve Growth Factor (NGF) and Brain-derived Growth Factor (BDNF) that signal through the receptors TrkA and TrkB, respectively. NGF and BDNF can have effects locally on axon terminals or signal retrogradely from the axon terminal to the cell body and induce changes in gene expression. NGF/BDNF long-range retrograde signaling is required for survival of sympathetic neurons, where retrograde transport of active endosomal signaling complexes promotes transcription of pro-survival/anti-apoptotic factors (*Riccio et al., 1997*; *Watson et al., 1999*; *Ye et al., 2003*). In addition to survival, retrograde transport of NGF/TrkA promotes sensory axon outgrowth by inducing transcriptional changes in Serum Response Factor (*Wickramasinghe et al., 2008*), whereas retrograde axonal transport of BDNF/TrkB is required for BDNF-mediated dendritic growth in cortical neurons (*Zhou et al., 2012*). In general, however, the transcriptional targets downstream of retrograde neurotrophic factor signaling that promote axon growth are still largely unknown.

The neurotrophic factor receptor 'REarranged during Transfection' (Ret) and its major ligand, Glial cell line-Derived Neurotrophic Factor (GDNF), have established roles in axon growth, enteric nervous system and kidney development, and cancer progression. Ret receptor has two primary expressed isoforms, Ret9 and Ret51, each with a unique amino acid sequence of the intracellular C-terminus. The two isoforms have different in vivo signaling properties and roles during development. Initial reports indicated Ret9 is required for proper enteric nervous system and kidney development, while Ret51 is dispensable for these processes in murine and zebrafish models (*de Graaff et al., 2001*; *Heanue and Pachnis, 2008*; *Wong et al., 2005*). However, later reports demonstrated

an important role for Ret51 in murine kidney and enteric nervous system development, suggesting that the two isoforms are partially redundant in these processes (*Jain et al., 2006*; *Jain et al., 2010*). These isoforms are also differentially trafficked and susceptible to degradation in both neurons and non-neuronal cells (*Richardson et al., 2012*; *Tsui and Pierchala, 2010*). In sympathetic neuron culture, RET9 degrades more slowly than RET51, which is thought to allow retrograde trafficking of RET9 and ultimately promote neuronal survival (*Tsui and Pierchala, 2010*). In human epithelial cell lines, RET51 is more frequently localized at the plasma membrane, recycled to the plasma membrane after endocytosis, and more rapidly internalized into endosomes after GDNF treatment than RET9 (*Crupi et al., 2015*; *Richardson et al., 2012*). Thus, Ret9 and Ret51 have notably different properties and responses to ligand binding even within the same cell.

Ret/GDNF signaling promotes sympathetic (*Enomoto et al., 2001*), sensory (*Honma et al., 2010*), and motor (*Bonanomi et al., 2012*) axon growth and pathfinding. Long-range retrograde RET/GDNF trafficking from distal axons is required for survival of sensory (DRG) neurons and mature sympathetic neurons in rat primary cell cultures (*Coulpier and Ibáñez, 2004*; *Tsui and Pierchala, 2010*). In sympathetic neurons, retrograde transport of GDNF ligand from axon terminals to cell bodies has been observed (*Coulpier and Ibáñez, 2004*; *Tomac et al., 1995*) but, thus far, retrograde transport of GDNF/Ret in sensory neurons has not. Ret/GDNF signaling can also promote sympathetic axon growth in vitro; however, retrograde signaling/transport is not required for this role over short periods of time (*Bodmer et al., 2011*; *Tsui and Pierchala, 2010*). Long-range GDNF/Ret signaling is required for motor neuron extension in developing mouse embryos where loss of GDNF ligand expression in target muscles leads to reduced neuronal expression of *Pea3* transcription factor and disrupts neuronal cell body positioning and dendrite patterning (*Haase et al., 2002*; *Vrieseling and Arber, 2006*). However, the mechanisms mediating Ret/GDNF retrograde transport and the full extent of the Ret/GDNF downstream transcriptional response mediating axon growth are unknown and in vivo studies of Ret retrograde transport and signaling are sorely lacking (*Ito and Enomoto, 2016*).

While Ret/GDNF is required for axon extension during development, whether it has a specific role in pioneer axon outgrowth is unknown. Pioneer neurons are the first to extend axons to a particular region or target, acting as a guide and scaffold for 'follower' axons. Pioneer neurons are important in the developing CNS and PNS for the initial navigation to appropriate targets, proper follower axon pathfinding, and promoting follower axon outgrowth. Despite their unique role in neurodevelopment, the molecular program regulates their behavior is not well known.

In vivo study of long-range neurotrophic factor signaling, receptor transport, and pioneer axon growth is technically challenging. Zebrafish embryos offer a number of advantages for these studies, including optical accessibility, rapid embryonic development, and amenability to transgenesis and genetic manipulation. In this study, we use the long sensory axons of the zebrafish posterior lateral line ganglion (pLLG) neurons because their planar character, superficial localization, and relatively rapid pioneer axon outgrowth make them uniquely suited for live imaging. The pLL senses water movement through mechanical stimulation of sensory hair cells that transduce this input through the pLL axons innervating them. During extension, 3 to 6 pLL pioneer axons are guided by the pLL primordium (pLLP), a group of cells that migrate along the trunk from 22 to 48 hr post-fertilization (hpf). Pioneer growth cones typically have longer and more elaborate filopodial protrusions compared to followers (*Bak and Fraser, 2003*; *Kim et al., 1991*) and this is the case in pLLG pioneer growth cones, which associate extensively with the migrating pLLP (*Sato et al., 2010*). The pLL primordium expresses GDNF and this ligand production is required for proper pLLG axon outgrowth, but not pLLG neuron specification or survival (*Schuster et al., 2010*). This allowed us to explore the specific mechanisms of long-range Ret-mediated axon outgrowth in sensory pioneer neurons.

We find that Ret is highly expressed in pioneer neurons and *ret51*, but not *ret9*, isoform is required for pioneer axon outgrowth. In the absence of Ret, pioneer axons display reduced growth cone volume and fewer filopodia. We also show that JNK-interacting protein 3 (Jip3) is required for retrograde transport of activated Ret51 from pioneer growth cones. In *jip3* mutants, retrograde transport of Ret51 is reduced, leading to accumulation of activated Ret in pioneer growth cones. Additionally, we observe live retrograde, but not anterograde, co-transport of tagged Ret51 and Jip3 in pioneer sensory axons. Finally, we find *myosin-X* (*myo10*), a regulator of filopodial formation, is downregulated in *ret* mutant sensory neurons. Myo10 protein is reduced in *ret* mutant pioneer growth cones, and loss of Myo10 produces axon extension defects similar to those in *ret* mutants.

This work describes a novel requirement for Ret51 in axon growth and pioneer neuron development, identifies a novel regulator of phosphorylated Ret51 retrograde transport, and suggests long-range retrograde Ret signaling regulates downstream transcriptional targets which are required for pioneer sensory axon growth cone dynamics.

## Results

### Ret expression is elevated in pioneer neurons during axon extension

Previous studies showed that *ret* is expressed in the pLLG neurons and the Ret ligand GDNF, which is expressed by the pLLP, is required for pLLG axon extension but not pLLG neuronal survival (*Schuster et al., 2010*). Because Ret receptor isoforms Ret9 and Ret51 have distinct properties, we asked which isoform is required for sensory axon extension. We assayed expression with a *ret9*-specific probe and a probe targeting a shared region of the *ret9* and *ret51* transcript (denoted *ret9+51*) in the pLLG during pioneer axon outgrowth (30 hr post-fertilization – hpf) by fluorescence in situ hybridization. Both *ret9* and *ret9+51* were expressed in the pLLG, with higher levels of both probes detected in the medial portion of the ganglion (*Figure 1A,B*, see *Figure 1—figure supplement 1*, panel A for probe target design).

Based on elevated levels of *ret* mRNA expression in the medial region of the ganglion, which was previously attributed to the location of pioneer neuron cell bodies (*Pujol-Martí et al., 2010*), we tested whether Ret protein was enriched in extending pioneer neurons. To label cell bodies of the extending pioneer pLLG axons, tails of 40 hpf *TgBAC(neurod:EGFP)$^{nl1}$* transgenic zebrafish embryos (hereafter referred to as *neurod:EGFP*) (*Obholzer et al., 2008*) were clipped just proximal to the pioneer axon terminals with rhodamine dextran-soaked scissors (*Figure 1C*). After 3 hr, embryos were imaged live to note the location of rhodamine-labeled cell bodies in pLLG, then fixed and immunostained with α-Ret primary antibody (for α-Ret antibody validation, see *Figure 1—figure supplement 1*, panel C). We found that the majority of rhodamine dextran labeled pioneer axons also displayed high levels of Ret protein (*Figure 1D',E'*). The dorsal localization of the cell bodies positive for the *ret9+51* probe in the same location of the ganglion strongly suggest that Ret receptor is highly expressed in pioneer neurons.

### The Ret51 isoform is required for lateral line axon extension

Because Ret is elevated in pioneer neurons during extension, we asked whether Ret activity is required for pioneer axon outgrowth, and, if so, which Ret isoform is necessary for this process. Using the *neurod:EGFP* transgene to visualize the pLLG axons, we observed that mutants homozygous for the null *ret$^{hu2846}$* allele (*Knight et al., 2011*) (which leads to loss of Ret9 and Ret51 isoforms) had disrupted pLLG axon extension, ultimately producing prematurely truncated axons (*Figure 1F, G*). We observed 97.4% of homozygous *ret$^{hu2846}$* mutants had premature truncation in one or both pLLG axons by three dpf (n = 89), although we also observed variability in truncation severity between embryos derived from individual adult mating pairs. However, within a group of embryos derived from a single mating pair, *ret$^{hu2846}$* phenotypes were consistent, as evidenced by the small standard error of the mean of axon truncation in *ret$^{hu2846}$* mutants in subsequent rescue experiments (*Figure 1H,I*). Because of pair to pair variability, in our subsequent experiments we compared *ret$^{hu2846}$* mutant embryos to sibling controls from the same single mating pairs. To test if a particular Ret isoform was required for pLLG axon outgrowth, we injected *ret$^{hu2846}$* mutant and sibling embryos with in vitro transcribed human Ret isoform mRNA tagged with mCherry, either *RET9-mCherry* or *RET51-mCherry* (*Crupi et al., 2015*). Injected animals were assayed at three dpf for axon length, measuring the shortest pLLG nerve in the embryo as a conservative measure of axon extension rescue (*Figure 1H,I*). Comparing uninjected *ret$^{hu2846}$* mutant embryos to *RET* isoform mRNA-injected *ret$^{hu2846}$* mutant siblings we found that *RET51,* but not *RET9*, significantly ameliorated the pLLG axon truncation phenotype (*ret$^{hu2846}$* mutant uninjected mean body segment location of axon termination: 17.5 ± 1.7 vs. *ret$^{hu2846}$* mutant *RET9* mRNA injected: 18.1 ± 1.6, p=1.00 by one-way ANOVA with post-hoc Tukey test, see for further details; *ret$^{hu2846}$* mutant uninjected: 13.8 ± 0.7 vs. *ret$^{hu2846}$* mutant *RET51* injected: 18.8 ± 0.9, p<0.001). In some cases, *RET51* injection produced full-length, bilateral rescue of axon extension (*Figure 1I*, 9 of n = 87 embryos). This suggests that Ret51 is the critical isoform required for Ret-mediated pLLG axon outgrowth.

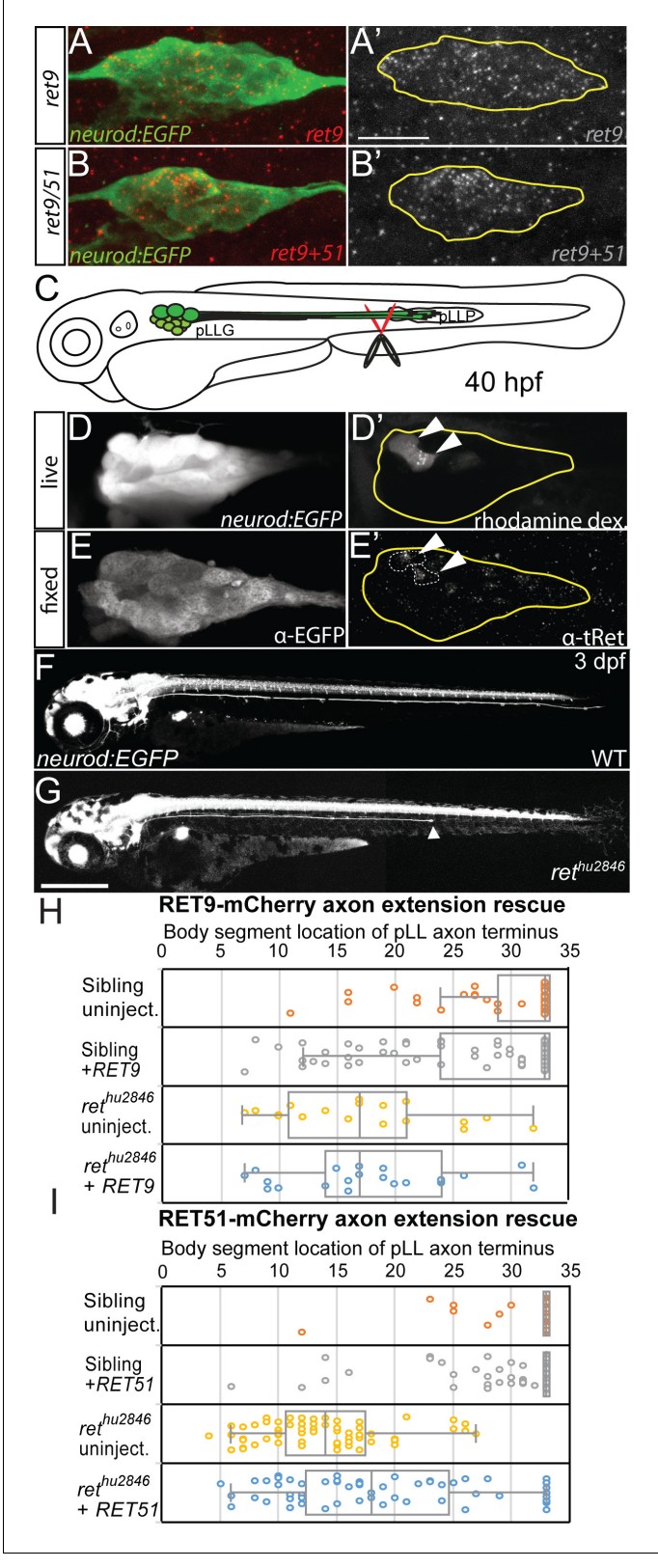

**Figure 1.** *ret* is highly expressed in pLL pioneer neurons and required for pLL axon outgrowth. (**A**,**B**) Lateral view of *ret9* and *ret9+51* probes, visualized by fluorescence in situ hybridization in *neurod:EGFP* embryos, reveals both are expressed mostly in dorsal cells during axon extension (30 hpf) in the pLLG (yellow line = pLLG outline, scale bar = 20 μm; probe design schematized in *Figure 1—figure supplement 1A*). (**C**) Diagram of pLL nerve sever

*Figure 1 continued on next page*

*Figure 1 continued*

experiment. Distal pioneer axons of *neurod:EGFP* transgenic embryos were severed during extension using rhodamine dextran soaked scissors. pLLP = posterior lateral line primordium. Live image of EGFP (**D**) and rhodamine-labeled pLLG pioneer neuron cell bodies (**D'**) 3 hr post-sever. Immunostaining for EGFP (**E**) and α-total Ret (**E'**, tRet) shows enriched Ret protein present in the rhodamine-positive cells (arrowheads) (8 of 10 cells analyzed from n = 5 embryos; Ret antibody validated in *Figure 1—figure supplement 1B,C*). (**F,G**) Live lateral view of wild-type siblings (**F**) and *ret^{hu2846}* mutants (**G**) with neurons labeled by the *neurod:EGFP* transgene. *ret^{hu2846}* mutants display truncated pLLG axons (arrowhead; scale bar = 200 μm; axon length and pioneer growth cone number over time displayed in *Figure 1—figure supplement 2A–D*). (**H,I**) Whisker plots comparing mRNA rescue of pLLG axon truncation defects within the same clutches of *ret^{hu2846}* mutants with injection of *RET9-mCherry* (**H**) or *RET51-mCherry* (**I**). *RET9-mCherry* injection did not significantly ameliorate *ret* mutant pLLG axon truncation (p=1.00 by one-way ANOVA with post-hoc Tukey test; sibling uninjected n = 50, sibling + *RET9* n = 98, *ret^{hu2846}* uninjected n = 18, *ret^{hu2846}* + *RET9* n = 21); in contrast, *RET51-mCherry* significantly increased mean pLLG axon length as well as cases of bilateral, full-length axon extension (p<0.001; sibling uninjected n = 70, sibling + *RET51* n = 166, *ret^{hu2846}* uninjected n = 61, *ret^{hu2846}* + *RET51* n = 87).

DOI: https://doi.org/10.7554/eLife.46092.002

The following source data and figure supplements are available for figure 1:

**Source data 1.** Quantification of *ret* mutant axon truncation rescue with *RET9* or *RET51* mRNA.
DOI: https://doi.org/10.7554/eLife.46092.005

**Figure supplement 1.** *ret* mRNA probe design and antibody validation.
DOI: https://doi.org/10.7554/eLife.46092.003

**Figure supplement 2.** *ret^{hu2846}* mutants lose distal pioneer growth cones over time but do not display axon retraction.
DOI: https://doi.org/10.7554/eLife.46092.004

## Ret is necessary for proper pioneer growth cone morphology

Because pLLG axon extension is incomplete in *ret^{hu2846}* mutants, we examined the cellular bases of this axon growth failure. After initial axon outgrowth and premature termination of axon extension in *ret^{hu2846}* mutant embryos, we observed no degeneration or retraction of the partially extended pLLG axons from 48 hpf through 120 hpf (*Figure 1—figure supplement 2*, panel A). To determine if there was a difference in the number of pioneer neurons that extended axons during initial outgrowth, we immunostained *ret^{hu2846}* mutants and their wild-type siblings with α-SCG10 antibody at 24 (early outgrowth) or 30 hpf (mid-outgrowth) and imaged distal pLLG axons. SCG10 (STMN2) is enriched in extending sensory pioneer growth cones (*Drerup et al., 2016*), serving as a marker of individual pioneer growth cones. Based on SCG10 expression in the distal 100 μm of axons (visualized by the *neurod:EGFP* transgene), the number of pioneer growth cones that initially extended from the pLLG at 24 hpf was not significantly different between *ret^{hu2846}* mutants and siblings (*Figure 1—figure supplement 2*, panel C, wild type: 5.73 ± 0.29 vs. *ret^{hu2846}*: 5.56 ± 0.48, p=0.99 by Mann-Whitney U test), indicating there is no defect in pioneer neuron specification or initial axon outgrowth in *ret^{hu2846}* mutants. However, at 30 hpf, *ret^{hu2846}* mutants had significantly fewer growth cones in the distal 100 μm of the growing pioneer axon bundle (*Figure 1—figure supplement 2*, panel D, wild type: 6.58 ± 0.32 vs. *ret^{hu2846}*: 4.00 ± 0.44, p<0.001). These observations suggest that pLLG pioneer neurons are specified and begin to extend normally; however, axon extension fails after initial outgrowth.

To explore the effect of Ret loss-of-function on pioneer growth cone morphology and behavior, we assayed axon terminals of extending pioneer axon bundles in *ret^{hu2846}* mutants and wild-type siblings (*Figure 2A,B*). First, we quantified axonal volume of the distal 75 μm of *neurod:EGFP* transgenic embryos at 30 hpf which typically contains 5–6 pioneer growth cones (*Figure 2C*). In comparison to siblings, *ret^{hu2846}* mutants had significantly reduced distal axonal volume in extending pioneer axon bundles (*Figure 2C*; wild type: 1477.6 ± 48.9 μm³ vs *ret^{hu2846}*: 1137.4 ± 136.1, p=0.003). To visualize individual pioneer axon and growth cone morphology, *neurod:EGFP* embryos at the one-cell stage were injected with plasmid containing the *neurod5kb* promoter (*Mo and Nicolson, 2011*) driving mCherry (*neurod5kb:mCherry*). Injected plasmid distributes mosaically in zebrafish embryos, allowing identification and live imaging of individual pLLG pioneer growth cones expressing mCherry during extension (30 hpf) (*Figure 2A',B'*). Compared to wild-type siblings,

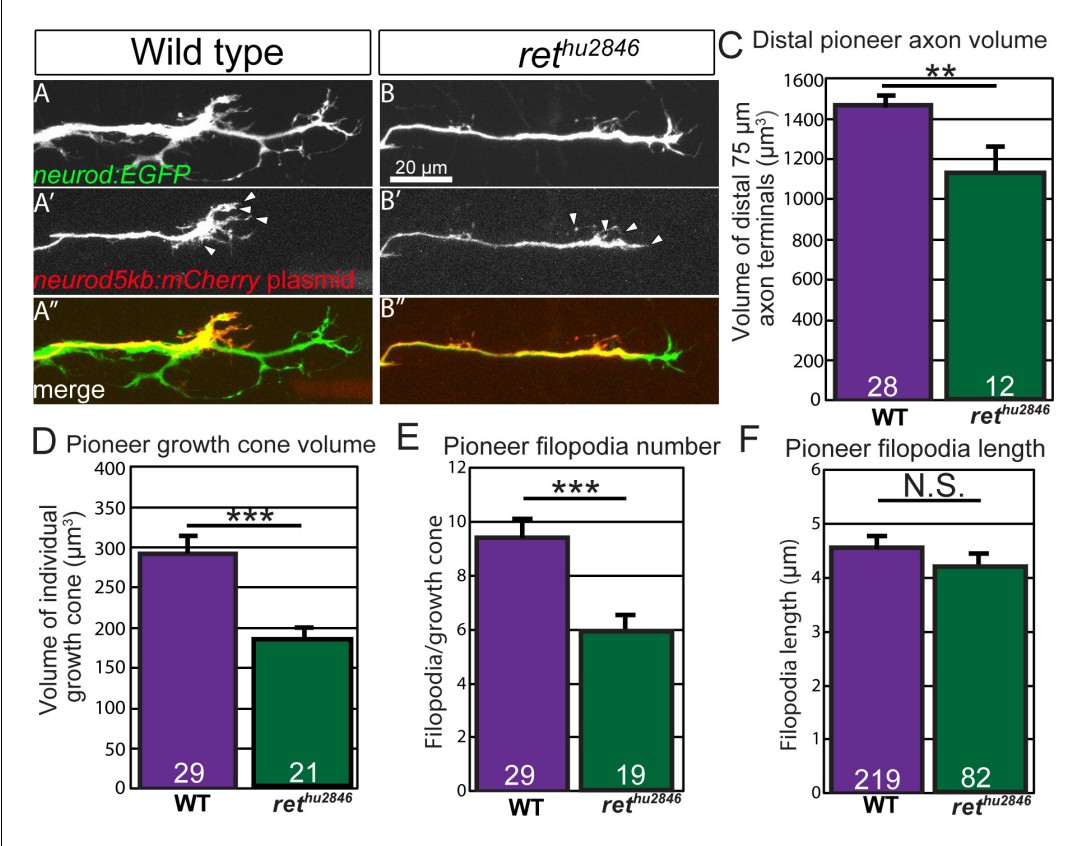

**Figure 2.** Ret loss-of-function alters pioneer axon growth cone morphology and protrusion. (A,B) Lateral view, live image of collective pioneer axon terminals of wild-type siblings (A) and $ret^{hu2846}$ mutants (B) labeled by *neurod:EGFP* transgene at 30 hpf. (A',B') Individual pioneer axons labeled by mosaic expression of *neurod5kb:mCherry* plasmid (Arrowheads = examples of counted filopodia). (C,D) Quantification of distal collective axon terminal volume (C) or individual pioneer growth cone volume (D). $ret^{hu2846}$ mutants have significantly reduced collective axonal and individual growth cone volume. (E,F) Quantification of number (E) and length (F, n = filopodia counted) of filopodia ($\geq$1 µm) per individual pioneer axon. Note that $ret^{hu2846}$ mutants have significantly reduced number of filopodia but not length. Error bars represent S.E.M.,**=p < 0.01, ***=p < 0.001, N.S. = not significant.
DOI: https://doi.org/10.7554/eLife.46092.006

The following source data is available for figure 2:

**Source data 1.** Individual embryo scoring data for scoring of growth cone and filopodia metrics in *ret* mutants.
DOI: https://doi.org/10.7554/eLife.46092.007

$ret^{hu2846}$ mutant pioneer growth cones were much less elaborate, had significantly reduced volume (**Figure 2D**; wild type: 291.5 ± 24.2 µm³ vs. $ret^{hu2846}$: 182.6 ± 17.4, p<0.001), and significantly fewer filopodia (defined as protrusions $\geq$ 1 µm) per growth cone (**Figure 2E**, wild type: 9.4 ± 0.7 vs. $ret^{hu2846}$: 5.9 ± 0.6, p<0.001). However, the length of these filopodia was not significantly different (**Figure 2F**, wild type: 4.55 ± 0.22 µm vs. $ret^{hu2846}$: 4.08 ± 0.29, p=0.20) between the two groups. Together, these data show that Ret signaling is required for maintaining proper pioneer growth cone morphology and filopodial numbers but not length.

## Jip3 loss-of-function phenocopies loss of Ret signaling

Neurotrophic factor signaling often involves retrograde transport of the activated receptor-ligand complex from the axon terminal to the cell body. However, the specific trafficking mechanisms are not clearly understood. In the pLLG, JNK-interacting protein 3 (Jip3) functions as a specific retrograde cargo adaptor for Dynein-mediated axonal transport during development (***Drerup and Nechiporuk, 2013***). Embryos homozygous for the Jip3 null allele, $jip3^{nl7}$, display pLLG axon extension defects similar to $ret^{hu2846}$ mutants (**Figure 3B**) (***Drerup and Nechiporuk, 2013***). Compared to siblings, $jip3^{nl7}$ mutant pLL nerves were significantly truncated by three dpf (**Figure 3—figure**

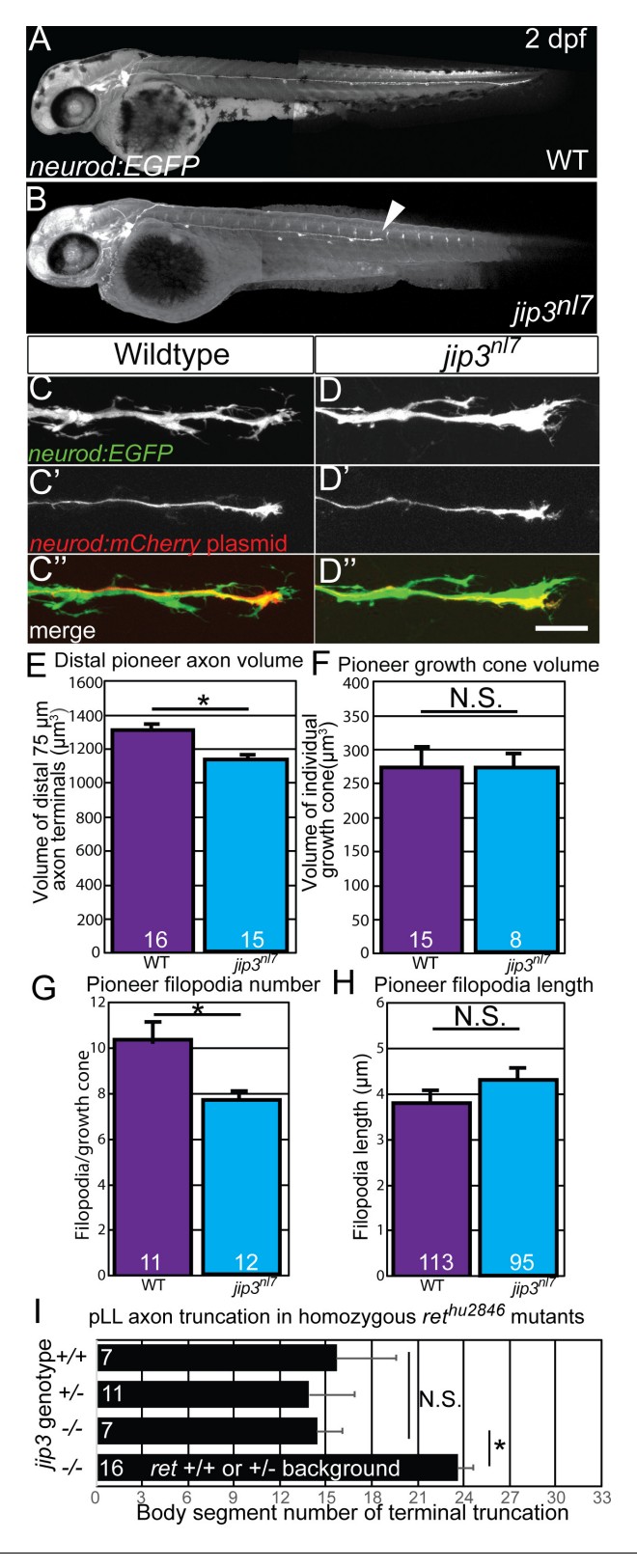

**Figure 3.** Loss of Jip3 phenocopies *ret* mutant pioneer axon defects. (**A,B**) Lateral view, live image of wild-type sibling (**A**) and *jip3^{nl7}* mutant (**B**) containing the *neurod:EGFP* transgene at two dpf. *Jip3^{nl7}* mutants display pLLG axon truncation defects (arrowhead; mutant truncation phenotype plotted in *Figure 3—figure supplement 1*). (**C, D**) Lateral view, live image of collective pioneer axon terminals of wild-type siblings (**C**) and *jip3^{nl7}* mutants (**D**)

*Figure 3 continued on next page*

*Figure 3 continued*

labeled by *neurod:EGFP* transgene at 30 hpf. Scale bar = 20 μm (C′,D′) Individual pioneer axons in the same animal labeled by mosaic expression of *neurod5kb:mCherr* plasmid. (E,F) Quantification of distal collective axon terminal volume (E) or individual pioneer growth cone volume (F). *Jip3^nl7* mutants have significantly reduced collective axonal volume but not individual growth cone volume. (G,H) Quantification of number (G) and length (H) of filopodia (≥1 μm) per individual pioneer axon. *Jip3^nl7* mutants have significantly reduced number of filopodia but not length. (I) Quantification of axon truncation defects of sibling, heterozygous, and homozygous *jip3^nl7* mutants in homozygous *ret^hu2846* background shows no significant difference in axon truncation phenotype between genotypes. *Jip3^nl7* homozygous mutants with +/+ or +/ret^hu2846* backgound had significantly less severe axon truncations. Error bars represent S.E.M., *=p < 0.05, N.S. = not significant.

DOI: https://doi.org/10.7554/eLife.46092.008

The following source data and figure supplement are available for figure 3:

**Source data 1.** Individual embryo scoring data for scoring of growth cone and filopodia metrics in *jip3* mutants and axon truncation in *jip3/ret* double mutants.
DOI: https://doi.org/10.7554/eLife.46092.010

**Figure supplement 1.** Quantification of *jip3^nl7* homozygous mutant axon truncation defects.
DOI: https://doi.org/10.7554/eLife.46092.009

---

*supplement 1*; mean body segment location of termination sibling: 33 ± 0 vs. *jip3* mutants: 24.0 ± 0.6; p<<0.0001 by Mann-Whitney U test). Thus, we investigated whether Jip3 and Ret act in the same pathway during axon outgrowth. We discovered that, similar to *ret^hu284* mutants, *jip3^nl7* mutants displayed reduced terminal collective pioneer axon volume at 30 hpf (*Figure 3D,E*, wild type: $1294.7 \pm 51.5$ μm$^3$ vs *jip3^nl7*: $1134.7 \pm 57.2$, p=0.024) and fewer filopodia per pioneer growth cone (*Figure 3G*, wild type: $10.3 \pm 0.9$ vs. *jip3^nl7*: $7.7 \pm 0.4$, p=0.017) with no significant change in filopodial length (*Figure 3H*, wild type: $3.88 \pm 0.38$ μm$^3$ vs. *jip3^nl7* mutant: $4.24 \pm 0.26$, p=0.33). Additionally, double homozygous *ret^hu2846* mutant and *jip3^nl7* mutants had no significant difference in axon truncation defects compared to single homozygous *ret* mutants (*Figure 3I*, body segment location of axon termination in *ret^hu2846* mutants, +/+: $14.4 \pm 1.7$, *jip3^nl7*/+: $13.9 \pm 3.0$, *jip3^nl7*/*jip3^nl7*: $15.7 \pm 3.9$, p=0.92 by one-way ANOVA), indicating Jip3 and Ret operate in the same genetic pathway. Notably, wild-type *ret* siblings homozygous for *jip3^nl7* had a less severe axon truncation defect compared to the *jip3/ret* homozygous double mutants (*jip3^nl7*/*jip3^nl7*; +/+ or *ret^hu284*/+: nl7 ± 1.1; p=0.047 by post hoc Tukey test). Based on these findings, we investigated whether loss of Jip3 affects localization of Ret protein in the extending axons. *jip3^nl7* mutants displayed significant accumulation of both total and phospho-Y905 (activated) Ret receptor (referred to as tRet and pRet, respectively; for α-Ret antibody validation, see *Figure 1—figure supplement 1*, panels B-E) at the pioneer growth cone during extension compared to wild-type siblings (*Figure 4A–F*, for tRet, wild type fluorescence intensity: $1.97 \pm 1.15$ vs. *jip3^nl7*: $45.09 \pm 12.44$, p<0.001; for pRet, wild type: $7.59 \pm 1.60$ vs. *jip3^nl7*: $20.54 \pm 3.90$, p=0.033). These data suggest that a pool of activated Ret receptor fails to be trafficked from the extending growth cone in *jip3^nl7* mutants.

We next asked whether retrograde movement of Jip3 is necessary for the clearance of Ret from the growth cone and axon extension. To test this, we injected EGFP-tagged mRNAs encoding full-length Jip3 (control), Jip3 lacking the p150 binding domain that is unable to interact with the Dynein motor complex (*Cavalli et al., 2005*) (*jip3Δp150*) or Jip3 lacking the JNK interacting domain (*jip3ΔJNK*) (*Morfini et al., 2009*) into one-cell stage embryos from a heterozygous +/*jip3^nl7* mutant incross (*Figure 4G*). Embryos were fixed at 3 days and immunostained for neurofilaments (3A10) to visualize pLLG axons and the proportion of embryos with axon truncation were scored. Expression of a full-length *jip3* completely rescued axon truncation at 3 dpf (*Figure 4G*, uninjected truncation rate: 17.4%, n = 23 vs. *jip3*: 0%, n = 29). Within embryos from a single mating pair, *jip3ΔJNK* also rescued truncation while *jip3Δp150* failed to rescue (uninjected: 15.8%, n = 38, *jip3ΔJNK*: 0%, n = 15, *jip3Δp150*: 22.2%, n = 36). Additionally, expression of full-length *jip3* mRNA rescued accumulation of pRet in pioneer growth cones (*Figure 4I*) but *jip3Δp150* did not (*Figure 4J*). Our previous work showed that loss of Jip3 does not affect JNK signaling in extending pioneer pLLG axon growth cones. Similarly, we found no significant difference in pJNK levels in growth cones of *ret^hu2846* mutants and wild-type siblings at 30 hpf (*Figure 4—figure supplement 1*). Altogether, these data indicate that the Jip3 association with the retrograde motor is required for Jip3-

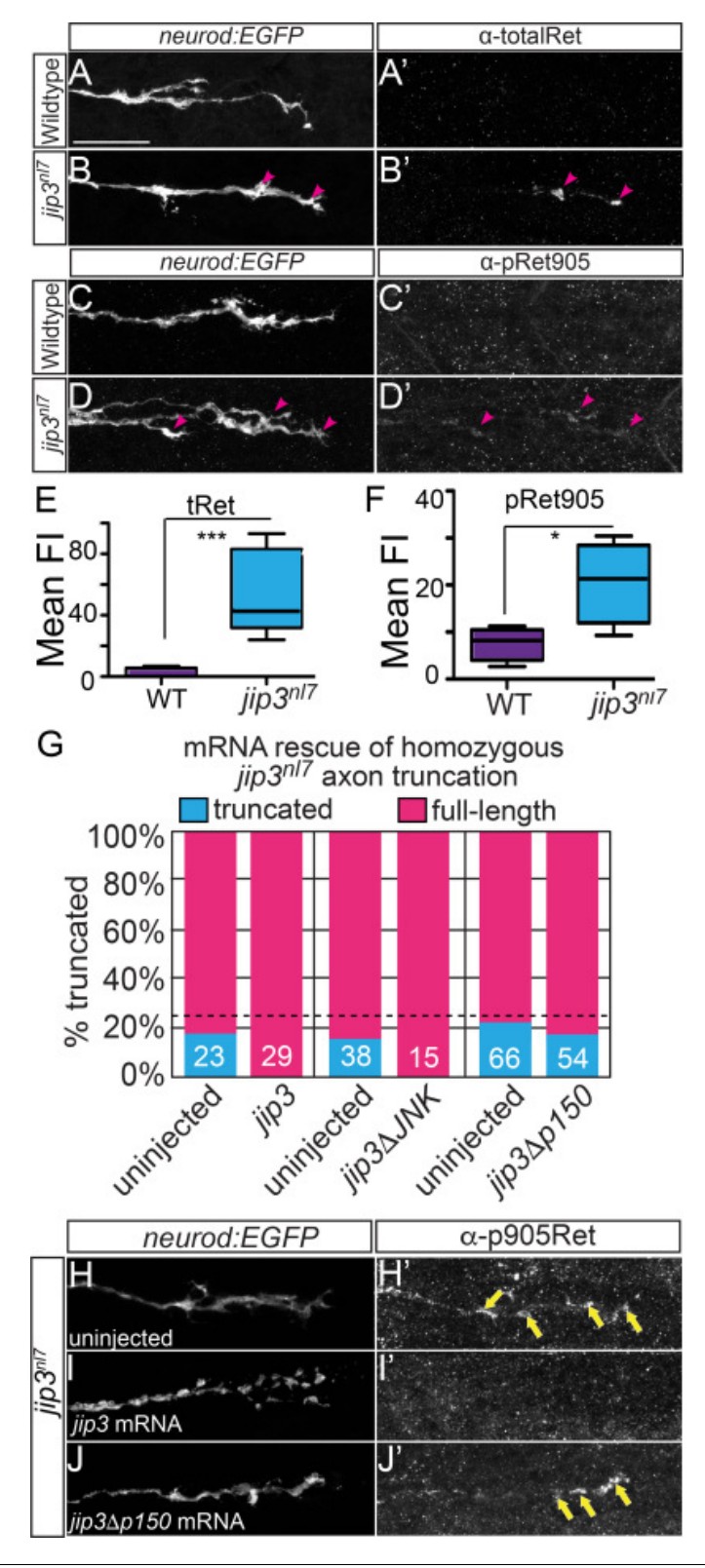

**Figure 4.** Jip3 is required for clearance of activated Ret from pioneer axon growth cones. (**A–D**) Lateral view, immunostained pioneer axon growth cones labeled by *neurod:EGFP* transgene at 30 hpf. Immunostaining for total Ret receptor (A',B', WT n = 8, *jip3^{nl7}* n = 5) and activated pRet905 receptor (C',D', WT n = 4, *jip3^{nl7}* n = 4) shows no signal in wild-type sibling pioneer growth cones (**A',C'**) but a notable accumulation of total and pRet905
*Figure 4 continued on next page*

*Figure 4 continued*

receptor in $jip3^{nl7}$ mutants (**B',D'**). Scale bar = 30 µm. (**E,F**) Quantification of immunostaining signal in pioneer growth cones displayed in whisker plots. Both total (tRet) and pRet905 signal is significantly higher in $jip3^{nl7}$ mutant pioneer growth cones. *=p < 0.05, ***=p < 0.001. (**G**) Embryos derived from $jip3^{nl7}/+$ heterozygous crosses were injected with wild-type *jip3, jip3* lacking the JNK-binding domain (*jip3ΔJNK*), or *jip3* lacking the p150-binding (*jip3Δp150*) mRNA. Expression of wild-type and *jip3ΔJNK* mRNA completely rescued pLLG axon truncation defects but *jip3Δp150* did not (phosph-JNK staining in *ret* mutants quantified in **Figure 4—figure supplement 1**). Dotted line is drawn at 25%, expected fraction of axon truncation in a heterozygous incross. Numbers represent *n* embryos. (**H–I**) Fixed and immunostained $jip3^{nl7}$ growth cones at 30 hpf from embryos injected with *jip3* mRNA constructs. Full-length *jip3* mRNA injection (**I**) rescues pRet accumulated in uninjected pLLG growth cones (**H**, yellow arrows) but *jip3Δp150* (**J**) does not.
DOI: https://doi.org/10.7554/eLife.46092.011
The following source data and figure supplement are available for figure 4:

**Source data 1.** Ret/phospho-Ret immunostaining quantification and *jip3* mutant construct rescue.
DOI: https://doi.org/10.7554/eLife.46092.013
**Figure supplement 1.** Anti-pJNK antibody immunostaining of pioneer growth cones.
DOI: https://doi.org/10.7554/eLife.46092.012

dependent axon extension and clearance of pRet from pLLG growth cones, suggesting disrupted Ret retrograde transport may underlie the axon truncation observed in $jip3^{nl7}$ mutants.

## Jip3 mediates retrograde transport of activated Ret51 receptor

Based on the role of Jip3 as a retrograde transport adapter and the similarity of $ret^{hu2846}$ and $jip3^{nl7}$ mutant axonal phenotypes, we asked if Jip3 mediates retrograde transport of Ret. To address this question, we first performed nerve sever experiments (**Drerup and Nechiporuk, 2013**) as an indirect readout of Ret axonal transport (**Figure 5A–F**). Post nerve sever, anterogradely transported cargoes accumulate in the proximal stump, whereas retrogradely transported cargoes accumulate in the distal stump (**Drerup and Nechiporuk, 2013**). We hypothesized that if Jip3 specifically mediates retrograde axonal transport of Ret from the growth cone, Ret protein should accumulate at the distal site of nerve sever. Extended pLLG pioneer axons (labeled by *neurod:EGFP*) of siblings and $jip3^{nl7}$ mutants were cut between neuromast 2 and 3 at five dpf, allowed to recover for 3 hr, fixed, and immunostained for tRet and pRet. Total Ret was detected at the proximal and distal cut sites, indicating that Ret is bidirectionally trafficked in axons. However, the amount of total Ret was not different between wild-type and $jip3^{nl7}$ mutant embryos (**Figure 5A–D**, proximal fluorescence intensity, wild type: 32.0 ± 2.5 vs. $jip3^{nl7}$: 28.4 ± 3.5, p=0.39; distal FI, wild type: 26.2 ± 2.3 vs. $jip3^{nl7}$: 28.4 ± 2.8, p=0.54), arguing that Jip3 is not required for this process. In wild-type siblings and $jip3^{nl7}$ mutant embryos, we observed no accumulation of pRet at the proximal stump (**Figure 5C,F**, wild type: 9.0 ± 2.0 vs. $jip3^{nl7}$: 9.8 ± 3.1, p=0.82); however, pRet accumulated significantly at the distal stump of wild-type siblings (**Figure 5C'**) compared to $jip3^{nl7}$ mutants, which had much less pRet accumulation at the distal cut site (**Figure 5D',F**, wild type: 22.4 ± 2.9 vs. $jip3^{nl7}$: 11.4 ± 1.8, p=0.004). Altogether, these data imply that Jip3 is specifically required for retrograde transport of activated Ret.

Next, we tested directly if Jip3 is required for axonal transport of Ret receptor by observing trafficking of Ret fusion protein in extending pLLG axons. To achieve this, we injected a plasmid driving mCherry-tagged RET51 under the *neurod5kb* promoter (*neurod5kb:RET51-mCherry*) at the one-cell stage and imaged RET51 transport in extending pLLG axons at 30 hpf (**Drerup and Nechiporuk, 2016**). Injected plasmid distributes mosaically in the developing embryo, thus, we can select embryos expressing fusion constructs in 1–3 pLLG neurons to assay Ret trafficking in individual axons during extension. We observed anterograde and retrograde Ret trafficking in embryos injected with *neurod5kb:RET51-mCherry* or *ret51-EGFP* (**Figure 5—figure supplement 1**). To test whether Jip3 regulates Ret51 retrograde axonal transport in pioneer neurons, we assayed *RET51-mCherry* transport in $jip3^{nl7}$ embryos and wild-type siblings. For our experiments, we used expression of the *neurod:EGFP* transgene to identify the most distal pLLG pioneer neurons also expressing RET51-mCherry (**Figure 5G,H**). Based on kymograph analysis, we found significantly fewer retrogradely trafficked RET51-mCherry particles in $jip3^{nl7}$ mutant pLLG pioneer neurons compared to wild-type

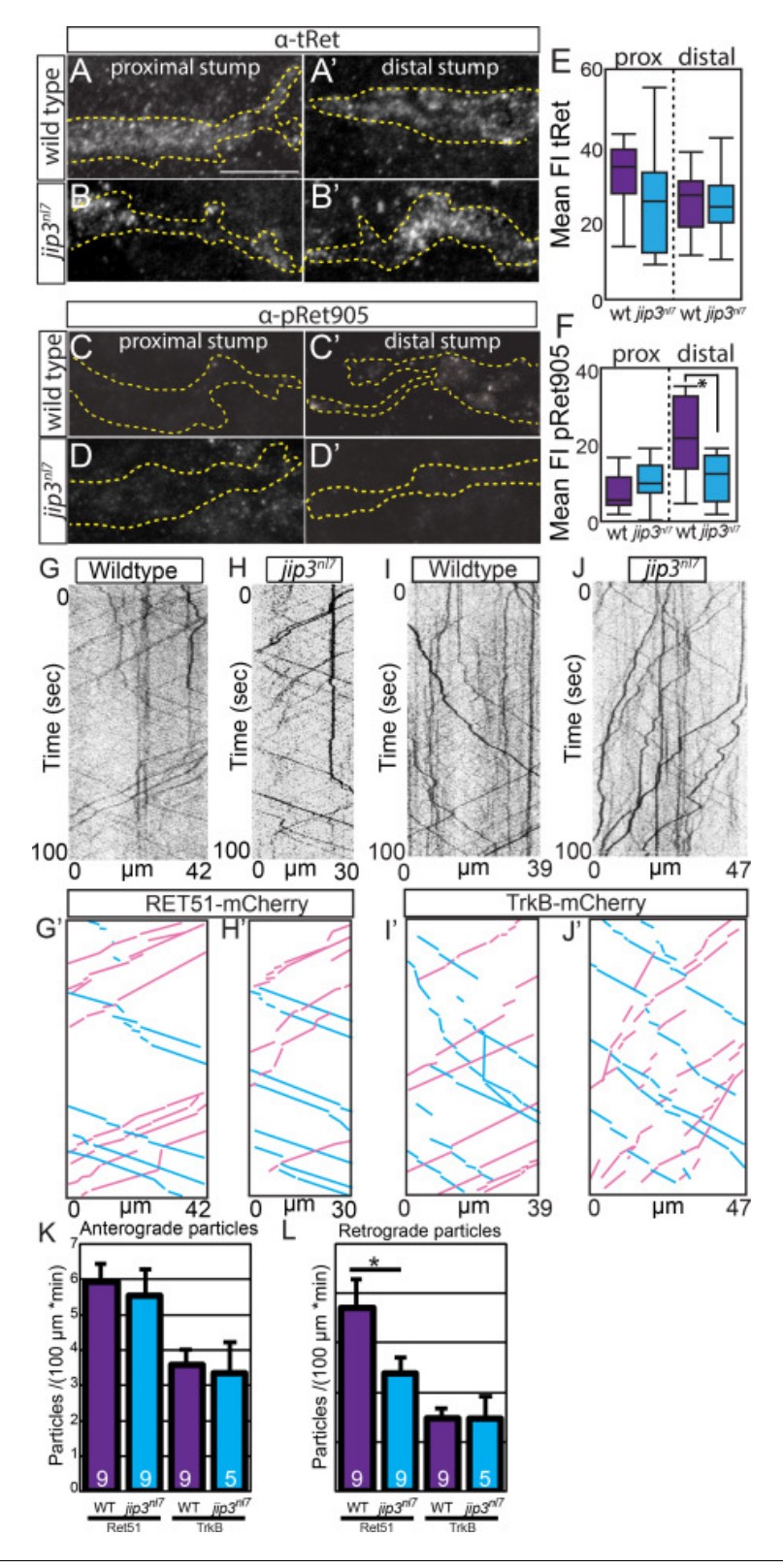

**Figure 5.** Jip3 mediates retrograde axonal transport of activated Ret51. Lateral view, immunostaining of severed pLL nerves proximal (**A–D**) and distal (**A'–D'**) to injury site, 3 hr post-injury; dotted line = nerve outline. (**A,B**) tRet immunostaining is not notably different between wild-type siblings (**A,A'**) and *jip3^{nl7}* mutants (**B,B'**) on either side of injury site. (**C,D**) p905Ret staining is mostly absent in wild-type siblings proximal to injury site (**C**) and *jip3^{nl7}* mutants on both sides of the injury site (**D,D'**). Scale bar = 10 µm. Stronger p905Ret staining signal is present in wild-type siblings distal to the injury

*Figure 5 continued on next page*

Figure 5 continued

site (C'), indicating presence of retrograde transport of activated Ret receptor in the axon. (E,F) Quantification of tRet (E; WT n = 12, *jip3*[nl7] n = 14) and p905Ret (F; WT n = 12, *jip3*[nl7] n = 12) staining following axon sever displayed by whisker plot (*=p < 0.05). Note p905Ret signal in wild-type siblings is significantly higher than in *jip3*[nl7] mutants, suggesting a failure of retrograde transport of pRet receptor in mutants. (G–J) Kymographs of trafficking in individual pLLG axons of Ret51-mCherry (G,H; Ret51 construct trafficking further shown in *Figure 5—figure supplement 1*) and TrkB-mCherry fusions (I,J) in wild-type siblings and *jip3*[nl7] mutants. (G'–J') Kymograph traces of scored anterograde (blue) and retrograde (magenta) transported particles. (K) Quantification of normalized anterograde particle counts from kymograph analysis (wild type Ret51 = 5.93 ± 0.52 vs. *jip3*[nl7] Ret51 = 5.53 ± 0.76, p=0.44 by Mann-Whitney U test; wild type TrkB = 3.58 ± 0.42 vs. *jip3*[nl7] TrkB = 3.36 ± 0.86, p=0.82). (L) Quantification of normalized retrograde particle counts from kymograph analysis (wild type Ret51 = 7.33 ± 1.17 vs. *jip3*[nl7] Ret51 = 4.73 ± 0.53, p=0.03; wild type TrkB = 3.07 ± 0.31 vs. *jip3*[nl7] TrkB = 3.06 ± 0.75, p=1.00). *jip3*[nl7] mutants show a significant decrease in the number of retrogradely transported Ret51-mCherry particles but no change in TrkB-mCherry particle counts or anterograde Ret51-mCherry particles. *=p < 0.05. Error bars represent S.E.M.

DOI: https://doi.org/10.7554/eLife.46092.014

The following source data and figure supplement are available for figure 5:

**Source data 1.** Quantification of immunostaining and kymograph analysis.
DOI: https://doi.org/10.7554/eLife.46092.016
**Figure supplement 1.** In vivo imaging of Ret51 trafficking.
DOI: https://doi.org/10.7554/eLife.46092.015

---

siblings, suggesting a disruption in retrograde transport of Ret51 (*Figure 5G',H',L*). To examine whether transport of other neurotrophic factor receptors expressed in the pLLG, such as TrkB (*Gasanov et al., 2015*), was disrupted, we analyzed trafficking of TrkB-mCherry fusion from embryos injected with *neurod5kb:trkB-mCherry* plasmid (*Figure 5I,J*). Notably, we saw no significant change in trafficking of TrkB-mCherry particles (*Figure 5K,L*). These data, taken together with pRet accumulation in the distal stump of severed *jip3*[nl7] mutant pLLG axons, strongly argue that Jip3 is required for transport of activated Ret51, but not TrkB, in pLLG axons.

Finally, if Jip3 acts as a specific cargo adaptor for retrograde pRet51 transport, we expect that Jip3 and Ret51 would be co-transported in extending pLLG axons. To address this, we co-injected *Tg(BAC)Ret51-EGFP* and *neurod5kb:jip3-mCherry* (*Drerup and Nechiporuk, 2013*) plasmid at one-cell stage and imaged pLLG pioneer axons expressing both fusions at 30 hpf. Co-localization of Jip3-mCherry and Ret51-EGFP was observed in a subset of retrogradely trafficked particles but no cases of co-localization were observed for anterogradely transported particles (*Figure 6A,B*, *Figure 6—video 1*). Taken together, these data suggest Jip3 preferentially mediates retrograde transport of activated pRet51.

Because Jip3 mediates retrograde transport of pRet and *jip3*[nl7] mutants display axonal phenotypes that are similar to the *ret*[hu2846] mutant, we hypothesized that Jip3 mediates long-range Ret signaling which is required for pLL axon outgrowth. However, *jip3*[nl7] mutants typically display less severe pLL axon truncations than *ret*[hu2846] mutants (*Figures 1I* and *3I*, S3). We reasoned that this difference may be due to both local and long-range requirements for Ret signaling in pLL axons. To distinguish between the local and long-range signaling, we looked for ways to acutely inactivate Ret receptor in *jip3*[nl7] mutants. To achive this, we used a multi-kinase inhibitor, sorafenib, which blocks signaling of multiple receptor-tyrosine kinases including Ret, where it potently inhibits Ret activation (*Plaza-Menacho et al., 2007*). To confirm that sorafenib can inhibit Ret in pLL axons, we treated *ret*[hu284] mutant and wild-type sibling embryos with 5 μM sorafenib from 24 to 72 hpf (*Figure 6—figure supplement 1*). Sorafenib treatment of wild-type siblings induced pLLG axon defects similar to those found in untreated *ret* mutants. Furthermore, we observed no significant exacerbation of the axon truncation in *ret* mutants treated with sorafenib compared to either untreated *ret* mutants or sorafenib-treated siblings. This suggests that, in this context, sorafenib functions as a specific inhibitor of Ret signaling. We then treated *jip3* mutant and sibling larvae with sorafenib (*Figure 6C*). Drug treatment induced significantly more severe axon truncations in siblings compared to untreated *jip3* mutants (mean body segment location of axon termination in sorafenib-treated siblings: 16.1 ± 1.5 vs. untreated *jip3*[nl7] mutants: 22.5 ± 1.1, p<0.001 by one-way ANOVA with post-hoc Tukey test). In addition, sorafenib-treated *jip3* mutants displayed similar truncation severity to sorafenib-treated siblings (sorafenib-treated *jip3*[nl7] mutants: 15.4 ± 1.0, p=0.96). These results are consistent with the observed difference in axon truncation severity between *ret* and *jip3* mutants and suggest local Ret

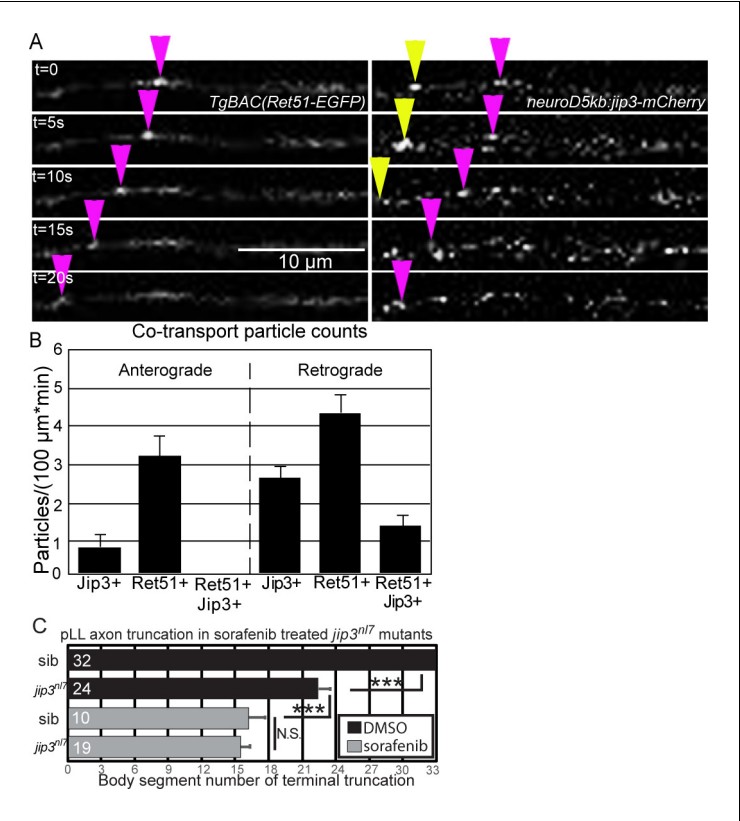

**Figure 6.** Ret51 and Jip3 co-transport in extending pLLG axons in vivo. (**A**) Still images from *Figure 6—video 1* of pLLG axonal transport in embryos co-injected with *neurod5kb:jip3-mCherry* and *Tg(BAC)Ret51-egfp* plasmids. Retrograde transport of mCherry+ eGFP+ particles can be seen (magenta arrowhead) as well as solely GFP+ particles (yellow arrowhead). (**B**) Quantification of Jip3-mCherry and Ret51-eGFP normalized particle counts (n = 5 embryos). Anterograde Ret51+ Jip3+ particles were not observed, however retrograde Ret51+ Jip3+ particles were seen. (**C**) Quantification of axon truncation defects of wild-type sibling and *jip3^{nl7}* mutant larvae at 72 hpf treated with DMSO or 5 μM sorafenib from 24 to 72 hpf (Sorafenib specificity confirmed with *ret* mutants in *Figure 6—figure supplement 1*). Sorafenib treatment produces a more severe truncation phenotype than the *jip3^{nl7}* mutation alone, but loss of Jip3 does not exacerbate the sorafenib-induced truncation phenotype. ***=p < 0.001, N.S. = not significant. Error bars represent S.E.M.

DOI: https://doi.org/10.7554/eLife.46092.017

The following video, source data, and figure supplement are available for figure 6:

**Source data 1.** Kymograph analysis of Jip3/Ret51 co-transport experiments.
DOI: https://doi.org/10.7554/eLife.46092.019
**Figure supplement 1.** Sorafenib induces pLLG axon truncation similar to that found in Ret genetic loss-of-function at 72 hpf.
DOI: https://doi.org/10.7554/eLife.46092.018
**Figure 6—video 1.** Time lapse image of Ret51-Jip3 co-transport.
DOI: https://doi.org/10.7554/eLife.46092.020

signaling in the growth cone promotes some aspects of axon outgrowth even in the absence of Jip3-mediated long-range retrograde transport.

## Ret signaling regulates transcription and pioneer growth cone levels of Myosin-X

Because retrograde transport is required for some aspects of Ret-mediated pLL sensory axon growth, we asked if there were specific transcriptional targets downstream of long-range Ret signaling that are required for pioneer axon outgrowth. Based on the observed defects in filopodial formation and growth cone advancement, we looked for factors that are specifically expressed in pLLG

neurons during axon extension and regulate filopodial dynamics. We focused on Myosin-X (Myo10), an unconventional Myosin that promotes initiation of filopodia and links integrins and the actin cytoskeleton to promote cell migration. Myo10 is upregulated during neurite growth following peripheral nerve injury (*Plantman et al., 2013*) and loss of Myo10 disrupts cortical axon growth (*Zhu et al., 2007*). Although humans and mice have one Myo10 gene, zebrafish have three with specific expression patterns. Of these, Myo10l1 has the most restricted embryonic expression and is the only one expressed in the pLLG (*Sittaramane and Chandrasekhar, 2008*). We hypothesized that loss of Ret signaling would reduce *myo10l1* transcription, leading to decreased Myo10 in pioneer growth cones. In turn, this would disrupt filopodia formation and subsequent axon growth in *ret* mutants. To test this, we analyzed *myo10l1* levels in *ret^hu2846* mutants and siblings at 30 hpf using fluorescence in situ hybridization (*Figure 7A,B*). *ret^hu2846* mutant ganglia had significantly reduced mean fluorescence intensity compared to siblings (*Figure 7B,C*). To verify if this reduced Myo10 protein in pioneer growth cones, we immunostained 30 hpf *ret^hu2846* mutants and sibling *neurod:EGFP* embryos with Myo10 antibody (*Figure 7D,E*). This antibody recognized My10l1-eGFP fusion protein expressed in zebrafish sensory neurons using BAC transgenesis (*Figure 7—figure supplement 1*, panel A). Consistent with the observed changes in *myo10l1* transcription, we found significantly reduced Myo10 immunostaining fluorescence in distal pioneer axons (*Figure 7E,F*). Similarly, the levels of Myo10l1 were reduced in aoxn growth cones of *jip3* mutants (*Figure 7G*). If Myo10l1 is required for proper pioneer axon outgrowth, we expect loss of Myo10l1 to disrupt pLLG axon growth, similar to loss of Ret signaling. To test this, we performed G0 CRISPR-Cas9-mediated genetic knockdown of *myo10l1* using the established method of injecting a multiplexed cocktail of three sgRNAs (*Shah et al., 2015*; *Wu et al., 2018*) that efficiently target three different regions of *myo10l1* in developing *neurod: EGFP* embryos (*Figure 7—figure supplement 1*, panels B,C). Overall, *myo10l1* crispant embryos displayed significant pLL nerve truncation compared to uninjected siblings (*Figure 7H*, crispant mean pLL truncation: $27.4 \pm 1.0$ vs. uninjected siblings: $32.9 \pm 0.1$, $p < 0.001$ by one-way ANOVA with post-hoc Tukey test, see Materials and methods for more details; n = 112 and 136, respectively), although pLL truncation was partially penetrant (n = 34 of 112 embryos). This suggests Myo10l1 is required for proper pioneer axon outgrowth and loss/reduction of Myo10l1 may contribute to *ret^hu2846* mutant axon truncation defects. Reduction of Myo10l1 in *ret^hu2846* mutants did not exacerbate the axon truncation phenotype ($17.0 \pm 1.2$ vs. *myo10l1* crispant *ret^hu2846* mutants: $17.1 \pm 1.7$, p=1.00; *Figure 7G*; n = 36 and 29, respectively), suggesting that Ret and Myo10l1 function in the same pathway.

## Discussion

While the ability of GDNF/Ret signaling to promote axon outgrowth during development is known, the mechanisms of long-range retrograde Ret signaling and consequent effects on transcription and pioneer axon behavior are largely unknown. Our work finds that the Ret51 isoform is required for sensory pioneer axon outgrowth. We observed enriched expression of *ret9+51* mRNA in a region of the pLLG known to hold the pioneer neuron cell bodies (*Pujol-Martí et al., 2010*). Through retrograde dye labeling of pioneer neuron cell bodies, we found an enrichment of Ret protein in pLLG pioneer neuron cell bodies, corresponding to the region of *ret9+51* mRNA expression. *ret^hu2846* mutants do not have defects in pLLG pioneer neuron specification or survival but do have a failure of continued growth of pioneer axons. This pioneer axon growth was rescued in *ret^hu2846* mutants by exogenous expression of *RET51* mRNA but not *RET9*, indicating that Ret51 is specifically required for pioneer axon outgrowth. These data also demonstrate that Ret is a specific marker of pioneer neurons in the pLLG.

We propose that long-range retrograde Ret signaling mediated by Jip3 is required for pioneer axon outgrowth and growth cone dynamics. While Ret/GDNF signaling could have local effects on growth cone behavior and morphology, we argue that long-range retrograde signaling and a transcriptional response underlie the loss of filopodia and a significant portion of the pLLG axon truncation phenotype described for several reasons. First, the Jip3 and Ret loss-of-function axon truncation and growth cone filopodia phenotypes are very similar. Second, there is no difference in the severity of axon truncation in *jip3^nl7/ret^hu2846* double mutants compared to individual *ret* homozygous mutants, suggesting they act in the same pathway. Third, in *jip3* mutants we observe an accumulation of phospho-Ret in extending growth cones and a reduction in retrograde axonal transport of

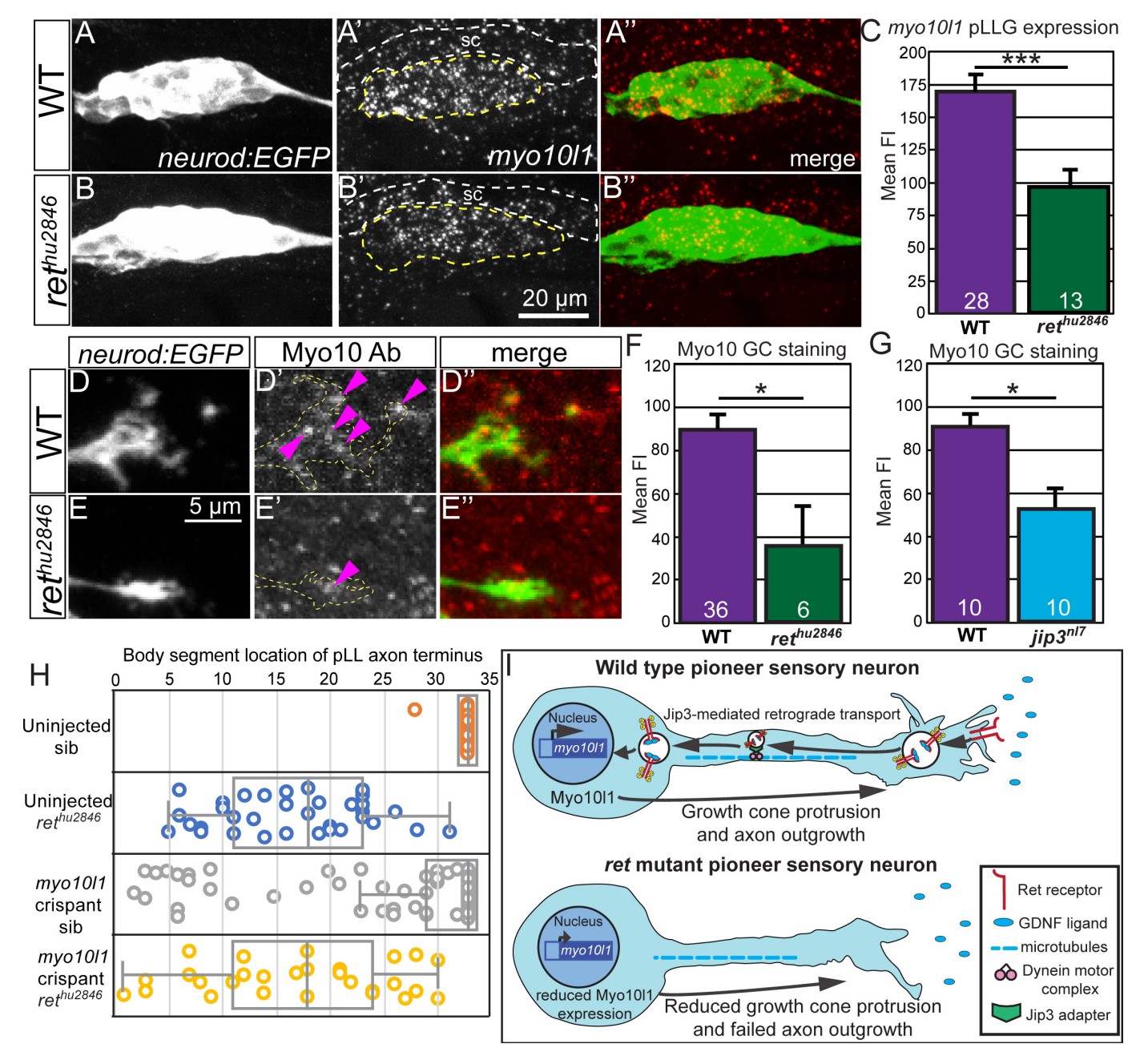

**Figure 7.** Myosin-X expression and growth cone localization are reduced in $ret^{hu2846}$ mutants. (A,B) Lateral view, pLL ganglia marked by *neurod:EGFP* of wild type and $ret^{hu2846}$ mutants at 30 hpf. (A',B') Fluorescent in situ hybridization shows decreased expression of *myosin-10l1* in *ret* mutants (yellow dotted line, ganglion; sc and white dotted line, spinal cord). (C) Quantification of mean fluorescence intensity of fluorescent in situ hybridization reveals significantly lower expression of *myosin-10l1* in *ret* mutant pLL ganglia. (D,E) pLL pioneer axon growth cones at 30hpf marked by *neurod:EGFP* expression. (D',E') Immunostaining for Myo10 shows a decrease in overall fluorescence and puncta (magenta arrowheads) in $ret^{hu2846}$ mutants (Antibody confirmation shown in ***Figure 7—figure supplement 1***, panel A). (F,G) Quantification of mean fluorescence intensity of Myo10 immunostaining in the distal 50 µm of pLL pioneer axons shows a significant decrease in Myo10 localization in $ret^{hu2846}$ (F) and $jip3^{nl7}$ mutants (G). (H) Whisker plot of sibling and *ret* mutant larvae (*neurod:EGFP*) injected with *myo10l1* gRNA/Cas9. *Myo10l1* CRISPR injection in wild-type *ret* siblings-induced axon truncation in many cases but did not significantly exacerbate axon truncation in *ret* mutants (*myo10l1* transcript and CRISPR targeting shown in ***Figure 7—figure supplement 1***, panels B,C). p. Error bars represent mean S.E.M., *=p < 0.05, ***=p < 0.001 (one-way ANOVA with post hoc Tukey test). (I) Proposed model of the contribution of long-range retrograde Ret signaling to axon outgrowth of pioneer sensory neurons.

DOI: https://doi.org/10.7554/eLife.46092.021

The following source data and figure supplement are available for figure 7:

*Figure 7 continued on next page*

*Figure 7 continued*

**Source data 1.** Quantification of FISH/immunostaining and scoring *ret* mutant/*myo10l1* crispant axon truncation.
DOI: https://doi.org/10.7554/eLife.46092.023
**Figure supplement 1.** Assessment of anti-Myo10 antibody and *myo10l1* gRNA efficiency.
DOI: https://doi.org/10.7554/eLife.46092.022

Ret51 and pRet by in vivo imaging and immunostaining following nerve sever, respectively. Fourth, *jip3* constructs lacking the p150 Dynein complex binding domain cannot rescue axon truncation or distal pRet accumulation in *jip3* mutants. However, *jip3* lacking the JNK binding domain fully rescues these defects, and we observe no changes in growth cone pJNK levels in *jip3* or *ret* mutants, indicating Jip3 retrograde transport is required for proper axon outgrowth and retrograde transport of pRet from extending growth cones. Finally, in *ret* mutants we observe a reduction in *myo10l1* expression and Myo10l1 in pioneer growth cones, suggesting a specific downstream transcriptional target of Ret signaling that underlies one of the observed growth cone defects. Consistent with this, knockdown of Myo10l1 disrupted pioneer axon extension similar to *ret^hu2846* mutants. Thus, we provide evidence for a mechanism by which long-range retrograde signaling induces a specific transcriptional change that in turn regulates pioneer growth cone dynamics and outgrowth. This model is consistent with previous observations in cultured sympathetic and sensory neuron indicating that neurotrophic factor-dependent axon growth over long periods of times requires a transcriptional response (*Bodmer et al., 2011*).

Our data also provide support that, in addition to a long-range Ret-mediated retrograde signaling, local Ret activity is necessary for pLL axon growth. We found that genetic loss of Jip3, which presumably blocks retrograde Ret signaling, produces a less severe axon truncation phenotype than loss of Ret. We also find that pharmacological inhibition of Ret in wild-type axons induces a more significant axon truncation than in untreated *jip3* mutants. Finally, we noted differences in the pLLG pioneer axons of *jip3* mutants compared to *ret* mutants. Specifically, unlike *ret* mutants, individual growth cone volume is not significantly reduced in *jip3* mutants. Overall, these observations argue that a combination of both local and long-range Ret signaling are required for axon outgrowth. In fact, we would like to suggest that Jip3-mediated retrograde transport may not be necessary for early stages of the pLLG pioneer axon outgrowth, but is required for continued axon growth as axons extend beyond a certain distance from the soma. This model is consistent with previous observations in cultured sympathetic and dorsal rood ganglia neurons, in which long-range retrograde transport of TrkA receptor is required for long-term (but not short-term) maintenance of axon outgrowth (*Bodmer et al., 2011*).

## Ret signaling regulates pioneer growth cone morphology and dynamics

Regulated growth cone dynamics, including formation and disassembly of filopodia and lamellipodia, is required for growth cone advancement and axon outgrowth. Loss of Ret led to fewer filopodia per pioneer growth cone but did not change mean length of filopodia. This suggests a defect in initiation of filopodial formation but not maintenance or continued growth. Consistent with this result, we observed reduction in Myo10l1, an unconventional Myosin that promotes filopodia formation, in *ret* mutants (*Bohil et al., 2006*; *Zhu et al., 2007*). In addition, transient CRISPR-based knockdown of Myo10l1 in *ret* mutants did not exacerbate the axon truncation phenotype observed in *ret* mutants alone, suggesting that Myo10l1 acts downstream of Ret. Although, to conclusively determine if *myo10l1* acts in a linear pathway with Ret in pLLG neurons during pioneer axon extension, germline *ret;myo10l1* double mutants carrying the *neurod:EGFP* transgene should be generated and their offspring analyzed for axon truncation defects. Disruption of dynamic filopodial formation can prevent proper growth cone motility (*Lowery and Van Vactor, 2009*), leading to failed axon outgrowth. Another non-mutually exclusive possibility is that failure to dynamically form filopodia could disrupt proper interaction with the pLLP (which produces GDNF and BDNF; *Gasanov et al., 2015*) necessary for guiding and promoting axon growth. In this case, less interaction with the pLLP could reduce reception of trophic/permissive cues produced by the pLLP, further reducing the formation of growth cone filopodia and producing a positive feedback cycle that ultimately ends in failed growth cone advancement.

In addition to the loss of filopodia, we also observe a significant reduction in pioneer growth cone volume. The reduced growth cone volume and more compact morphology of *ret* mutant growth cones which lack the typical spread out 'footprint' of pioneer growth cones could be caused by defects in formation of lamellipodia, which would ultimately prevent proper growth cone migration and steering. While changes in *myo10* expression are consistent with the filopodial defects and axon outgrowth failure, local Ret signaling may promote lamellipodia formation or other specific growth cone behaviors.

## Jip3 is an adapter for transport of activated Ret

While there is evidence that long distance GDNF signaling is required for motor axon pathfinding (*Vrieseling and Arber, 2006*), to date there has been no in vivo observation of retrograde axonal trafficking of Ret receptor or description of the mechanisms that would underlie this transport in neurons (*Ito and Enomoto, 2016*). In this study, we provide compelling evidence that Jip3 functions as an adapter for the retrograde transport of activated Ret to promote axon outgrowth. We found that Jip3 loss-of-function partially phenocopied pLLG axon truncation and many of the growth cone defects we observed in *ret^{hu2846}* mutants. We observed no difference in axon truncation defects in *jip3^{nl7}* and *ret^{hu2846}* null double mutants compared to *ret^{hu2846}* mutants, indicating that Ret and Jip3 act in the same pathway to promote axon growth. Our data also show that loss of Jip3 leads to accumulation of activated Ret in pioneer axon growth cones. The Jip3-mediated clearance of pRet requires Jip3 binding to the retrograde motor, consistent with the idea that Jip3 functions as a retrograde adapter in this context. Axon sever experiments showed that activated Ret is retrogradely transported in axons and Jip3 is required for that process. Consistent with this observation, we found a significant reduction in the number of retrogradely trafficked Ret51 particles in *jip3^{nl7}* mutants. Notably, we did not observe complete loss of retrograde transport of Ret51 in *jip3^{nl7}* mutants or observe complete co-localization of Jip3/Ret51 retrograde transport. This is likely due to the inability to distinguish between phosphorylated or non-activated tagged Ret receptor in vivo. These observations are consistent with our assumption that Jip3 is required for retrograde transport of activated Ret51, which presumably constitutes only a subset of Ret51 fusion visualized in our live trafficking experiments. Finally, we observed live retrograde co-transport of Jip3 and Ret51 tagged particles in extending pLLG axons, but no co-transport of anterogradely trafficked particles. Collectively, these results indicate Jip3 mediates retrograde transport of phosphorylated Ret51 in pioneer axons.

It is possible that Jip3 mediates local Ret signaling in pioneer growth cones, since it also can function as a scaffold that regulates local JNK phosphorylation. However, while Jip3 interaction with JNK has been previously implicated in axon extension (*Sun et al., 2013*), it is not necessary for pLL axon extension (*Drerup and Nechiporuk, 2013*) and we observe no change in pJNK levels in pioneer growth cones in *ret^{hu2846}* mutants. Consistent with this, we found that mRNA injection of full-length Jip3 and Jip3 lacking the JNK-binding domain in *jip3^{nl7}* mutants completely rescued axon truncation and accumulation of pRet in pioneer growth cones. Combined with our previous results, this indicates Jip3-mediated retrograde axonal transport of phosphorylated Ret51 is required for axon outgrowth and proper formation of growth cone filopodia.

## A specific role for Ret51 isoform in pioneer axon outgrowth

Ret receptor isoforms have different roles and requirements during development. Ret9 previously was described as the critical isoform in mouse kidney organogenesis, enteric nervous system development, and embryogenesis in general (*de Graaff et al., 2001*; *Wong et al., 2005*). However, later murine studies argue these isoforms have largely redundant developmental function (*Jain et al., 2006*; *Jain et al., 2010*). Ret isoform roles are little studied in zebrafish, though initial work indicated Ret9, but not Ret51, is required for proper enteric nervous system development (*Heanue and Pachnis, 2008*). Neuronal studies examining differences in Ret9 and Ret51 subcellular localization and degradation have focused on their roles in neuronal survival (*Tsui and Pierchala, 2010*). RET51 is expressed in a specific subset of RET9-expressing mouse olfactory neurons during embryonic development (*Kaplinovsky and Cunningham, 2011*), but the functional consequence of this is unknown. Our work establishes a novel, specific role for Ret51 in pioneer axon extension unrelated to neuronal differentiation or survival and describes in vivo observation of axonal Ret51 isoform trafficking.

Though Ret9 failed to rescue mutant axon truncation, it may be that Ret9 has a role in local signaling at the dendrites or cell body that is unrelated to axon outgrowth.

These isoform-specific functional differences may be explained by differential trafficking properties and degradation rates of Ret isoforms. In cultured mammalian cells (HEK293 and SH-SY5Y cells), RET51 is preferentially localized to the plasma membrane, more rapidly internalized upon ligand binding and is differentially ubiquitylated compared to RET9 (*Crupi et al., 2015*; *Hyndman et al., 2017*). Different rates of Ret51 degradation are also seen in distinct neuronal populations. In mouse sympathetic neuron culture, RET51 is rapidly degraded in axon terminals following GDNF treatment (*Tsui and Pierchala, 2010*). However, in DRG sensory neuron cultures, RET51 was substantially more resistant to degradation and activated RET51 persisted much longer than in sympathetic neurons, which is correlated with detection of pRET51 in neuron cell bodies. Thus, it is possible that the plasma membrane localization and rapid internalization properties of Ret51 combined with the enhanced longevity of pRet51 in sensory neurons allow for its unique role in pioneer sensory neurons and, potentially, the ability to undergo long-range retrograde axonal transport.

In summary, our in vivo data demonstrate a novel role for Ret51 in pioneer axon extension and suggest the requirement of long-range, retrograde Ret signaling in this process. Additionally, we establish a system to examine the role of neurotrophic factor signaling in pioneer axon growth without a confounding role in neuronal survival. We identify that Jip3 mediates Ret51 retrograde transport, providing a useful tool to dissect local versus long-range signaling contributions of Ret receptor. Combined with our observation that Ret is specifically and highly expressed in pLLG pioneer sensory neurons, we submit that Ret51 functions as a molecular and transcriptional marker of these pioneers. This finding opens up several new technical avenues to examine the mechanisms underlying pioneer neuron development and pathfinding. In particular, this finding can be leveraged with transcriptomic techniques, such as single-cell RNA-seq, to study the transcriptional mechanisms that specify pioneer neurons and promote their enhanced axon growth and pathfinding capabilities. Finally, our results offer a holistic mechanism of long-range retrograde axonal transport of a neurotrophic factor receptor (Ret) inducing a specific transcriptional change (*myo10l1*) that ultimately impacts local protein levels in the growth cone (Myo10l1) required for proper axon behavior during development (filopodia formation and axon extension, *Figure 7I*). By elucidating the specific roles of concrete components involved in this process, our mechanistic account conceptually advance the paradigm of long-range neurotrophic factor signaling in axon growth. Better understanding of the fundamental processes that regulate the unique axonal growth and pathfinding capabilities of pioneer neurons can provide potential therapeutic points of intervention to promote axonal regeneration in response to injury or disease. Future studies will focus on identifying additional potential transcriptional targets downstream of Ret retrograde transport and how these regulate specific changes in pioneer growth cone morphology and behavior.

## Materials and methods

### Key resources table

| Reagent type (species) or resource | Designation | Source or reference | Identifiers | Additional information |
|---|---|---|---|---|
| Strain (*Danio rerio*) | *TgBAC(neurod:EGFP)$^{nl1}$; neurod:EGFP* | DOI: 10.1523/JNEUROSCI.5230–07.2008 | | |
| Strain (*Danio rerio*) | *ret$^{hu2846}$* | DOI: 10.1242/dev.061002 | ZIRC Catalog ID: ZL3218 | |
| Strain (*Danio rerio*) | *jip3$^{nl7}$* | DOI: 10.1371/journal.pgen.1003303 | | |
| Antibody | SCG10(STMN2) antibody (rabbit polyclonal) | ProteinTech | Cat. No.: 10586-AP | 1:100 |
| Antibody | GFP (rabbit polyclonal) | Invitrogen | Cat. No.: A11122 | 1:1000 |
| Antibody | GFP (chick polyclonal) | Aves Labs | Cat. No.: GFP-1020 | 1:1500 |

*Continued on next page*

*Continued*

| Reagent type (species) or resource | Designation | Source or reference | Identifiers | Additional information |
|---|---|---|---|---|
| Antibody | Phospho-Ret (pTyr905) (rabbit polyclonal) | Invitrogen | Cat. No.: PA517761 | 1:50 |
| Antibody | Ret (rabbit polyclonal) | Santa Cruz Biotechnology | Cat. No.: sc-167 | 1:100 |
| Antibody | Neurofilament (mouse monoclonal) | Developmental Studies Hybridoma Bank | Designation: 3A10 | 1:500 |
| Antibody | Myosin-X (rabbit polyclonal) | ProteinTech | Cat. No.: 24565–1-AP | 1:500 |
| Antibody | Phospho-JNK (Thr183/Tyr185) (rabbit polyclonal) | Cell Signaling | Cat. No.: 9251S | 1:100 |
| Antibody | Goat anti-chicken Alexa Fluor 488 secondary antibody | Invitrogen | Cat. No.: A-11039 | 1:1000 |
| Antibody | Goat anti-rabbit Alexa Fluor 568 secondary antibody | Invitrogen | Cat. No.: A-11008 | 1:1000 |
| Antibody | Goat anti-mouse Alexa Fluor 568 secondary antibody | Invitrogen | Cat. No.: A-11004 | 1:1000 |
| Antibody | Cy5 Goat anti-mouse IgG secondary antibody | Jackson Immunoresearch Laboratories | Cat. No.: 115-175-003 | 1:1000 |
| Antibody | Goat anti-rabbit Alexa Fluor 647 secondary antibody | Jackson Immunoresearch Laboratories | Cat. No.: A-27040 | 1:1000 |
| Recombinant DNA reagent | pCDNA3.1-RET9-mCherry | DOI: 10.1111/tra.12315 | | |
| Recombinant DNA reagent | pCDNA3.1-RET51-mCherry | DOI: 10.1111/tra.12315 | | |
| Recombinant DNA reagent | *neurod5kb:mCherry* | This paper | | Gateway destination plasmid pDestTol2pA2, recombination described in 'Plasmids and tagged constructs' in Materialsand methods |
| Recombinant DNA reagent | *neurod5kb:RET51-mcherry* | This paper | | Gateway destination plasmid pDestTol2pA2, recombination described in 'Plasmids and tagged constructs' in Materialsand methods |
| Recombinant DNA reagent | *TgBAC(ret51-eGFP)* | This paper | | Contains DKEY-192P21 clone, with 49.89 kb upstream and 68.78 kb downstream sequence of *ret* |

*Continued on next page*

*Continued*

| Reagent type (species) or resource | Designation | Source or reference | Identifiers | Additional information |
|---|---|---|---|---|
| Recombinant DNA reagent | *TgBAC(myo10l1-eGFP)* | This paper | | Conatins clone DKEY-10E5, with 20.98 kb upstream and 48.7 kb downstream sequence of *myo10l1* |
| Chemical compound, drug | sorafenib | Cell Signaling | Cat. No.: 8705 | 5 µM in 0.5% DMSO in embryo medium |
| Software, algorithm | Imaris | Bitplane | v8.4.2 | |
| Software, algorithm | SPSS Statistics | IBM | v25 | |

## Zebrafish husbandry

Adult zebrafish were maintained at 28.5°C as previously described and embryos were derived from natural matings or in vitro fertilization (*Westerfield, 2000*), raised in embryo media, and developmentally staged (*Kimmel et al., 1995*). Strains utilized were AB, *TgBAC(neurod:EGFP)^{nl1}* (*Obholzer et al., 2008*), *ret^{hu2846}* (*Knight et al., 2011*), and *jip^{nl7}* (*Drerup and Nechiporuk, 2013*).

## Plasmids and tagged constructs

*Neurod5kb:jip3-mCherry* was previously described (*Drerup and Nechiporuk, 2013*). *Neurod5kb: mCherry* plasmid was generated by recombining the *neurod5kb* promoter (*Mo and Nicolson, 2011*) p5e entry vector into the pDEST394 vector in the Gateway system by previously described methods (*Kwan et al., 2007*). Human RET9-mCherry and RET51-mCherry constructs in the EGFP-N1 vector were obtained from the Mulligan lab (*Crupi et al., 2015*) and digested with HindIII, NotI, and SphI to release the whole tagged construct and digest the EGFP-N1 vector backbone. Released RET fusion construct cassettes were ligated into HindIII/NotI-digested Gateway pME-MCS vector and subsequently recombined with the *neurod5kb* or *cmv/sp6* promoter p5e entry vectors to yield *neurod5kb:RET9/51-mCherry* or *cmv/sp6:RET9/51-mCherry* in the pDEST394 destination vector (*Kwan et al., 2007*). Jip3 deletion constructs Δp150 and ΔJNK were derived from pME-Jip3 (*Drerup and Nechiporuk, 2013*) using the Quickchange II kit (Agilent) (primers in *Table 1*). mRNA was synthesized from Jip3 constructs or *cmv/sp6:RET9/51-mCherry* using SP6 mMessage Machine (Life Technologies) and microinjected at 500 pg/embryo for Jip3 constructs and 50 pg/embryo for RET9/51-mCherry constructs.

## CRISPR-Cas9-mediated knockout

Three sgRNAs for CRISPR-Cas9 knockdown of *myo10l1* were designed and injections were performed as previously described (*Shah et al., 2015*) (*Table 1*), with the modification that, instead of synthesizing and injecting individual sgRNAs, a cocktail of all three sgRNAs was amplified and synthesized simultaneously in one reaction (*Wu et al., 2018*), then injected in *neurod:EGFP* embryos. At 4 dpf, embryos with obvious necrosis or body morphology defects were removed and body segment location of pLLG axon terminals was scored for injected and uninjected embryos, and genomic DNA was isolated for individual embryos. CRISPR efficiency was evaluated by amplifying regions surrounding CRISPR cut sites (primers in *Table 1*) and incubating amplicons with restriction enzymes with cut sites adjacent to CRISPR targets (gRNA A: PstI; gRNA B: BslI; gRNA C: BstNI).

## Generation of the Ret51- and Myo10l1-eGFP BAC fusions

We modified Ret- and Myo10l1-containing bacterial artificial chromosome (BAC) clones by *Escherichia coli*-based homologous recombination (*Suster et al., 2011*). BAC clone DKEY-192P21 contains 49.89 kb of sequence upstream and 69.78 kb of sequence downstream of *ret*; BAC clone DKEY-10E5 contains 20.98 kb sequence upstream and 48.7 kb of sequence downstream of *myo10l1*

**Table 1.** PCR primers and oligonucleotides.

| | |
|---|---|
| Δp150forward | GGGAAAGAAGTGGAAAATGAGGAGCTGGAATCGGTA |
| Δp150reverse | TTTACCGATTCCAGCTCCTCATTTTCCACTTCTTTC |
| ΔJNKforward | GAGGAAAAGTAAAACAGGTGGAGATGGCATGGAGGA |
| ΔpJNKreverse | CCATGCCATCTCCACCTGTTTTACTTTTCCTCCAGT |
| CRISPR Myo10l1 A | aattaatacgactcactataGGTGGTGTACCTGCAGCAGG gttttagagctagaaatagc |
| CRISPR Myo10l1 B | aattaatacgactcactataGGAGGATACCCGCCAGATGG gttttagagctagaaatagc |
| CRISPR Myo10l1 C | aattaatacgactcactataGGACAAGAGTTCCTGGTCAG gttttagagctagaaatagc |
| Myo10l1 A primer F | TCCTCTCCCCTTTTGTGAAGTA |
| Myo10l1 A primer R | TGTCTTTCTGGTATCGCTGATG |
| Myo10l1 B primer F | AATCCTTTCAGAGTTGCAGACA |
| Myo10l1 B primer R | CGTGGACACACTGTCTTCTCTT |
| Myo10l1 C primer F | CTGTAACTCAGACCTGCCAGAA |
| Myo10l1 C primer R | CCTGTGTCACAAAAGCAACATT |
| Ret 3'RACE-d(T) | GACCACGCGTATCGATGTCGACTTTTTTTTTTTTTTTTV |
| Ret 3'RACE anchor | GACCACGCGTATCGATGTCGAC |
| Ret-3'RACE-F | CGAACCCTCCCCTCCACTTG |
| Ret9 unique 3'RACE F | AGAATTTCCCATGCATTTACTAGA |
| Ret9+51 3'RACE F | CGCTCAACAGACTCGATGCC |
| Ret9 unique probe R | CTGCTGCGGTGACATTGTAT |
| Ret51 BAC rec F | ggctttgcattctcccgcaaaaatcgtggaca caatcgatagtatggtgagcaagggcga |
| Ret51 BAC rec R | caccccccgtgtctttccgccattgattttggc ttgcgttttgcagaattcgcccttga |
| My10l1 BAC rec F | cgctgcagcatcatgtccatcaccagcaaca gcagcgcctggaatggtgagcaagggcga |
| My10l1 BAC rec R | ttgtgtagctgtacatttggagctctgggtggt gtcgatcacctgcagaattcgcccttg |

DOI: https://doi.org/10.7554/eLife.46092.024

(http://www.sanger.ac.uk/Projects/D_rerio/mapping.shtml). After recombination, the modified *ret* BAC clone contained an *eGFP* gene positioned in frame at the 3' end of exon 20 to generate Ret51 fusion, following SV40 polyadenylation signal and kanamycin resistance gene. The modified Myo10l1 BAC clone contained an *eGFP* gene positioned in frame at the 3' end of exon 40 to generate My10l1 fusion. The kanamycin resistance gene, flanked by FLP sites, was removed by FLP-mediated recombination (*Suster et al., 2011*). The accuracy of EGFP integration and FLP recombination were evaluated by PCR, sequencing, and by transient expression assays.

## In situ hybridization and wholemount immunostaining

Templates for digoxygenin-labeled antisense RNA probes were generated for the *ret9* specific region and *ret9+51* by amplifying the alternatively spliced 3' exons and 3' UTR from cDNA using 3' Rapid Amplification of cDNA Ends (3'-RACE) (*Scotto-Lavino et al., 2006*) with listed primers (*Table 1*). '3'RACE-d(T)' primer was used for the initial reverse transcription reaction to generate cDNA specifically. '3'Anchor R' primer specific to the end of 3'RACE-d(T) oligo was first used with a Ret-3'RACE-F primer to amplify both isoforms. This PCR product was subjected to a nested PCR with either 'Ret9-3'unique RACE F' or 'Ret9+51 RACE F' primers to amplify either a unique regions for *ret9* or a region containing both *ret9+51* probes, respectively, and cloned into pCR4-TOPO (Thermofisher). Fluorescence in situ hybridization was performed as previously described (*Jülich et al., 2005*) by digesting PCR4-TOPO clones with SpeI or NotI and digoxygenin-labeled

antisense RNA probe synthesized with T7 or T3 polymerase (Life Technologies). *myo10l1* digoxygenin-labeled antisense RNA probe was generated using an EST containing 4.5 kb of *myo10l1* (Dharmacon, #EDR1052-208859531), digesting with SmaI and synthesized with T7 polymerase (Life Technologies).

Whole mount immunohistochemistry was performed following established protocols (*Ungos et al., 2003*) with the following exception: embryos stained with anti-Ret were fixed in Shandon Glyo-Fixx (Thermo Scientific) for one hour at room temperature. The following antibodies were used: anti-GFP (1:1000; Invitrogen #A11122), anti-SCG10 (1:100; ProteinTech, #10586-AP) anti-p905Ret (1:50; Thermo Pierce, #PA517761), anti-Ret (1:100; Santa Cruz, #sc-167), anti-neurofilament (1:500; DSHB, 3A10), anti-Myo10 (1:500; Proteintech #24565–1-AP), anti-pJNK (1:100; Cell Signaling #9251S). tRet antibody was validated by verifying lack of immunofluorescence signal in pLLG in *ret*[hu2864] mutant embryos compared to wild-type siblings (*Figure 1—figure supplement 1*, panels B, C). pRet antibody was validated by injecting a plasmid construct encoding RET51-mCherry fusion (*neurod5kb:RET51-mCherry*), identifying embryos with 2–3 pLLG neurons expressing the construct, fixing at 30 hpf, and immunostaining. Co-localization of Ret51-mCherry and pRet was found in extending growth cones, but pRet was largely absent in the ganglion or cell bodies expressing Ret51-mCherry (*Figure 1—figure supplement 1*, panels D,E), consistent with expectation that only a subpopulation of Ret51 would be phosphorylated and that subpopulation would be enriched in extending growth cones near the source of GDNF ligand. For pioneer nerve sever experiments (*Figure 3C–E*), α-rabbit AlexaFluor 647 secondary (Thermo Fisher) was used to distinguish α-Ret antibody signal from rhodamine. All fluorescently labeled embryos were imaged using a 40X/NA = 1.3 oil objective on a FV1000 laser scanning confocal system (Olympus). Brightness and contrast were adjusted in Adobe Photoshop and figures were compiled in Adobe Illustrator.

## Quantification of immunofluorescence

For analysis of pRet and tRet intensity in axon terminals and after nerve injury, individuals were immunolabeled as described above. For consistency of labeling, compared larvae were processed in the same batch. Confocal Z-stacks (0.5 μm between planes) were taken of the area of interest using a 40X/NA = 1.3 oil objective with identical settings. Images were analyzed using ImageJ (*Abramoff et al., 2004*). For fluorescence intensity measurements of pRet or tRet in wild type and mutant growth cones, summed projections of the regions of interest were generated only through regions that contained the *neurod:EGFP* signal and converted to eight bit in ImageJ as previously described (*Drerup and Nechiporuk, 2013*). Briefly, in the pLL nerve injury analysis, a 30 μm, *neurod:EGFP*-positive region encompassing the proximal or distal edge of the severed axon was selected and summed projections through only this segment were compiled for analysis. Prior to statistical comparison, the mean background fluorescent intensity, measured in a region adjacent to the injury site, was subtracted from the values generated.

For Myo10 immunostaining and *myo10l1* fluorescent in situ hybridization, wholemount embryos were imaged as described above using 488 and 568 nm excitation channels. Myo10 or *myo10l1* signal intensity was measured in pLL ganglia and collective axon terminals using Imaris (Bitplane) as follows. We used EGFP signal to generate surfaces of either the ganglia or the distal 50 μm of pioneer neuron terminals based on full stacks. We measured 568 nm channel mean fluorescence intensity of axon terminals or ganglia and subtracted the average mean fluorescence intensity of three regions proximal to the ganglion/axon terminals to account for background staining.

## Growth cone imaging and analysis

For imaging, embryos were mounted in 1.5% low melting point agarose on a glass coverslip, submerged in embryo media containing 0.02% tricaine (MS-222; Sigma) and imaged at 24 or 30 hpf using a 60X/NA = 1.2 water objective on an upright Fluoview1000 confocal microscope (Olympus). The distal 75–125 μm of pLLG axon terminals, indicated by expression of *neurod:EGFP* transgene, were imaged with stacks of sufficient depth to capture all portions of the terminal axons. For individual growth cone imaging, embryos were injected with *neurod5kb:mCherry* plasmid, embryos expressing mCherry in 1–2 pLLG neurons with axons extending at or near the axon terminals (marked by *neurod:EGFP* expression) were selected and imaged as described using 488 and 568 nm excitation channels. Growth cone and collective axon terminal volume was quantified using Imaris

software (Bitplane) as described above. EGFP signal was used to generate axonal surfaces of individual growth cones or the distal 75 µm of collective axon terminals, and volume was measured in Imaris. Filopodia were visualized by creating Z-projections of image stacks of individual growth cones in ImageJ and filopodia were traced using the segmented line tool, measured for length, and filopodia $\geq$1 µm number and length were scored.

### Retrograde axon labeling experiment

40 hpf *neurod:EGFP* embryos were anesthetized and mounted in 1.2% low melt agarose. Tails were severed with rhodamine dextran-soaked scissors just rostral to pLLP growth cones. Embryos were freed from agarose, allowed to develop for 3 hr, then re-mounted and imaged using the FV1000 confocal microscope. Embryos were subsequently fixed and stained to detect Ret immunoreactivity.

### Axon transport analysis

Zygotes were injected with plasmid DNA encoding fluorescently tagged cargos of interest with expression driven by the *5kbneurod* promoter (*Mo and Nicolson, 2011*) and imaged as described above in 'growth cone imaging and analysis.' At 30 hpf, embryos were sorted under epifluorescence to identify individuals with tagged cargo expression in a few cells of the pLL ganglion. For each embryo, a region of interest (30–150 µm) was selected in the pLL nerve in which a long stretch of axon was observable in a single plane. Scans were taken at three frames per second for 500–1000 frames. Embryos were subsequently released from agarose and processed for genotyping. For co-transport, embryos expressing both constructs in a single cell were selected and imaged as described above using sequential imaging of the 488 and 568 nm excitation channels. 500 frames were collected at three frames per second.

Transport parameters were analyzed using kymograph analysis in the MetaMorph software package (Molecular Devices, Inc) as previously described (*Drerup and Nechiporuk, 2016*). Kymographs containing 10 or more traces were analyzed and each were averaged within individual embryos for statistical analysis. The number of particles moving in each direction was estimated based on traces on the kymographs and then normalized to length of axonal segment and total imaging time.

### Axotomy and image acquisition

Five-day old zebrafish larvae (*neurod:EGFP* carriers) were anesthetized in 0.02% tricaine and embedded in 3% methylcellulose on a slide. Pulled thick-walled glass capillaries were used to sever the nerve between NMs 2 and 3. Slides were immersed in Ringer's solution (116 mM NaCl, 2.9 mM KCl, 1.8 mM CaCl$_2$, 5 mM HEPES pH = 7.2, 1% Pen/Strep) and incubated at 28.5°C for 3 hr. Larvae were then collected and immunolabeled for pRet or tRet and EGFP.

### Sorafenib drug treatment

Sorafenib was dissolved in DMSO to make a 10 mM stock. Embryos were transferred at 24 hpf into embryo media containing 0.5% DMSO and 5 µM sorafenib. Embryos were placed into freshly prepared sorafenib/DMSO media at 48 hpf and then scored for axon truncation at 72 hpf.

### Statistical analysis

Data were analyzed with the SPSS software package (IBM). Data suitable for parametric analysis were analyzed using ANOVA with post-hoc Tukey test. Data not suitable for parametric analysis were analyzed using Mann-Whitney *U* test (Wilcoxon ranked sum test). Axon truncation length measured within groups over time (*Figure 1—figure supplement 2*, panel A) was analyzed by repeated measures ANOVA with post hoc Bonferroni test for pairwise comparison. For experiments involving two independent variables (rescue of *ret* mutant phenotype with *RET* isoform mRNA injection or injection of *myo10l1* gRNA/Cas9 in *ret* mutants and siblings) two-way ANOVA was performed to test if main effects and interactions were statistically significant. If interaction was statistically significant, ANOVA was repeated with simple effects, and significance of main effects was re-evaluated and post hoc tests were performed.

### Ethics statement

All animal works were approved by and conducted according to guidelines of the Oregon Health and Science University IACUC.

## Acknowledgements

We thank Dr. Lois Mulligan for her gift of *RET9-mCherry* and *RET51-mCherry* constructs and helpful discussion. Funding was provided to AVN from NICHD (1R01HD072844; http://www.nichd.nih.gov) ) and NINDS (1R01NS111419; http://www.ninds.nih.gov).

## Additional information

### Funding

| Funder | Grant reference number | Author |
|---|---|---|
| Eunice Kennedy Shriver National Institute of Child Health and Human Development | 1R01HD072844 | Alex V Nechiporuk |
| National Institute of Neurological Disorders and Stroke | 1R01NS111419 | Alex V Nechiporuk |

The funders had no role in study design, data collection and interpretation, or the decision to submit the work for publication.

### Author contributions

Adam Tuttle, Conceptualization, Formal analysis, Validation, Investigation, Visualization, Methodology, Writing—original draft, Project administration, Writing—review and editing; Catherine M Drerup, Validation, Investigation, Methodology, Writing—review and editing; Molly Marra, Investigation, Methodology, Writing—review and editing; Hillary McGraw, Investigation, Writing—review and editing; Alex V Nechiporuk, Conceptualization, Resources, Supervision, Funding acquisition, Validation, Investigation, Visualization, Methodology, Writing—original draft, Project administration, Writing—review and editing

### Author ORCIDs

Adam Tuttle (ID) https://orcid.org/0000-0002-4300-3848
Catherine M Drerup (ID) http://orcid.org/0000-0002-0219-3075
Alex V Nechiporuk (ID) https://orcid.org/0000-0002-8295-8188

### Ethics

Animal experimentation: This study was performed in strict accordance with the recommendations in the Guide for the Care and Use of Laboratory Animals of the National Institutes of Health. All of the animals were handled according to approved institutional animal care and use committee (IACUC) protocols of Oregon Health and Science University. The protocol was approved by the IACUC committee of the OHSU Research Integrity Office (Protocol ID: IP00000495). All live imaging and larval and adult procedures were performed under MS-222 anesthesia, and every effort was made to minimize suffering.

### Decision letter and Author response

Decision letter https://doi.org/10.7554/eLife.46092.028
Author response https://doi.org/10.7554/eLife.46092.029

## Additional files

### Supplementary files

• Transparent reporting form

DOI: https://doi.org/10.7554/eLife.46092.025

**Data availability**

Experimental data underlying Figures 1-7 have been uploaded with the manuscript for publishing as source data.

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
