## [Decision Letter]

Thank you for submitting your article "Retrograde Ret signaling controls sensory pioneer axon outgrowth" for consideration by *eLife*. Your article has been reviewed by three peer reviewers, and the evaluation has been overseen by a Reviewing Editor and Didier Stainier as the Senior Editor. The reviewers have opted to remain anonymous.

The reviewers have discussed the reviews with one another and the Reviewing Editor has drafted this decision to help you prepare a revised submission.

Summary:

In this study the authors present novel and rigorous data demonstrating that Ret, the receptor for the GDNF family of neurotrophic factors, is critical for axon extension and growth cone dynamics of sensory pioneer neurons of Zebrafish. This function requires Ret51 specifically, and not the shorter Ret9 splice variant. Interestingly, the retrograde transport of Ret51 is necessary for regulation of axon extension/growth cone dynamics, and requires Jip3 for retrograde transport. While Jip3 is best known as an adaptor protein critical for *JNK* signaling, in this context the authors demonstrate that its activity as a cargo adaptor for retrograde motors is required and not the *JNK* binding domain. The authors further demonstrate that the retrograde Ret signal upregulates the expression of Myosin-10l1, which is critical for filopodial growth in these neurons. Taken together this manuscript provides new mechanistic insight into retrograde Ret signaling and identifies a novel, surprising function for this retrograde signal: axon extension and growth cone dynamics. There are however some points that should be addressed before the paper can be accepted.

Essential revisions:

1) Jip3 mutant phenotype shown in Figure 3B should be quantified.

2) Differences in growth cone morphology in Jip3 mutants are difficult to appreciate (Figure 3C-D") and although statistically significant (Figure 3E-G), these changes are less substantial than those observed in RET mutants (Figure 2C-E). Could these changes reflect local RET signaling in the distal axon, while axon stalling is the result of both long-range and local RET signal activation?

3) Given the striking resemblance of the axon truncation phenotype of RET and Jip3 mutants and genetic evidence of a linear pathway, it was surprising to see a relatively limited co-transport of the two proteins (Figure 6). What is the reason of this apparent discrepancy? Does Jip3 account for all RET retrograde transport? Co-localization between Jip3+ particles and phosho-RET, which is only trafficked retrogradely (Figure 5), should be assessed.

4) The authors state: "Because retrograde transport is required for Ret-mediated pLL sensory axon growth…". Technically they haven't shown this, rather they've shown that retrograde transport is required for Jip3-mediated axon growth. To ascertain this, they would need to show that Ret51 cannot rescue the axon truncation phenotype when retrograde transport is blocked. This could be done by testing if Ret51 expression can rescue axon phenotype in the Jip3/Ret double mutant. This experiment would strengthen their conclusions and the double mutant appears to be in hand. However, if the results are difficult to interpret, an alternative would be to tone down the conclusions on this point

5) The specificity of Myo10Ab staining (Figure 7D'-E") should be confirmed in myo10I1 mutants. In addition, it would be interesting to determine whether Myo10 expression changes in the cell body and axons of Jip3 mutants with blunted RET long-range signaling.

6) The conclusion that retrograde Ret signaling and transcriptional activation of myo10 is based on quantification of fluorescence in situ hybridization intensity. This may not be the best approach because absolute fluorescence intensity can vary between embryos. Could the authors show a difference in mRNA levels with quantitative real-time PCR?

---

## [Author Response]

Essential revisions:1) Jip3 mutant phenotype shown in Figure 3B should be quantified.

As requested, we have completed quantification of the axon truncation in *jip3* homozygous mutants and wild-type siblings and included that data as Figure 3—figure supplement 1. Quantification of *jip3* mutant nerve truncation has also been added to a panel in Figure 6C.

2) Differences in growth cone morphology in Jip3 mutants are difficult to appreciate (Figure 3C-D") and although statistically significant (Figure 3E-G), these changes are less substantial than those observed in RET mutants (Figure 2C-E). Could these changes reflect local RET signaling in the distal axon, while axon stalling is the result of both long-range and local RET signal activation?

Indeed, the *jip3* mutant growth cone phenotype is less severe than that of the *ret* mutant. Moreover, the axon truncation is less severe in *jip3* mutants compared to *ret* mutants, further implying that Jip3 mediates only a subset of Ret activities in the extending axon. We agree with the reviewer that the differences between these two mutant phenotypes could be explained by Ret acting as both a long- and short-range signal and Jip3 mediating the long-range aspect of Ret signaling. To better address this question, we blocked Ret signaling in *jip3* mutants using a pharmacological approach. This resulted in an axon truncation phenotype comparable to the *ret* mutant, which is more severe than that of *jip3* alone (new Figure 6C). This experiment, together with additional data presented in the manuscript, is consistent with a model where Jip3 mediates long-range signaling, but is not required for local Ret signaling. To discuss this model we included the following paragraph in Discussion:

“Our data also provide support that, in addition to a long-range Ret-mediated retrograde signaling, local Ret activity is necessary for pLL axon growth.[…] This model is consistent with previous observation in cultured sympathetic and dorsal rood ganglia neurons, in which long-range retrograde transport of TrkA receptor is required for long-term (but not short-term) maintenance of the axon outgrowth (Bodmer et al., 2011).”

3) Given the striking resemblance of the axon truncation phenotype of RET and Jip3 mutants and genetic evidence of a linear pathway, it was surprising to see a relatively limited co-transport of the two proteins (Figure 6). What is the reason of this apparent discrepancy? Does Jip3 account for all RET retrograde transport? Co-localization between Jip3+ particles and phosho-RET, which is only trafficked retrogradely (Figure 5), should be assessed.

Actually, limited co-transport of Jip3 and Ret51 is not surprising, but is consistent with a model where Jip3 is only required for retrograde transport of the activated or pRet (concluded from accumulation of pRet but not tRet in axon sever experiments). However, in our co-transport experiments, the Ret51-mCherry fusion construct labels both tRet and pRet. We thank the reviewers for their suggestion of immunostaining for Jip3, Ret51, and phospho-Ret. This is an ideal experiment to test the above model. We attempted many different permutations of tagged constructs and immunostaining protocols, but ultimately, it was difficult to draw any conclusions about co-localization from the signal specificity and resolution we observed. We saw some examples that suggested that Jip3/Ret51/pRet co-localized (see Author response image 1), but in most cases we observed staining that did not resolve clearly enough to distinguish individual puncta for scoring (see Author response image 1). One possible reason for our technical difficulties is that, in our hands, pRet antibody only works with Glyofixx, a glyoxal based fixative that contains considerable amounts of alcohol. We presume this affects lipid-based axonal vesicles (including those containing Ret and Jip3), such that we could never get a robust, reproducible signal to quantify the colocalization. Thus, with this approach we are unable to make a reliable claim regarding the ratio of Ret/Jip3/pRet co-localization.

**Author response image 1. respfig1:** Triple antibody staining for Jip3-mcherry, Ret51-eGFP, and phospho-Ret (Y905). (**A**) We detected instances of axonal puncta that were triple positive (magenta arrowheads), as well as Jip3 and Ret51 particles that were not positive for the other markers (white arrowheads). However, in most cases (**B**), staining was variable and distinguishing puncta and reducing background were difficult to achieve. Note that in this case Jip3 and Ret51 signal is not localized to distinct puncta, as observed during live imaging.

Thus, we decided not to include this data in the revised manuscript, but did addressed this issue in our Discussion as follows:

“Notably, we did not observe complete loss of retrograde transport of Ret51 in *jip3^nl7^*mutants or observe complete co-localization of Jip3/Ret51 retrograde transport. This is likely due to inability to distinguish between phosphorylated or non-activated tagged Ret receptor in vivo.”

4) The authors state: "Because retrograde transport is required for Ret-mediated pLL sensory axon growth…". Technically they haven't shown this, rather they've shown that retrograde transport is required for Jip3-mediated axon growth. To ascertain this, they would need to show that Ret51 cannot rescue the axon truncation phenotype when retrograde transport is blocked. This could be done by testing if Ret51 expression can rescue axon phenotype in the Jip3/Ret double mutant. This experiment would strengthen their conclusions and the double mutant appears to be in hand. However, if the results are difficult to interpret, an alternative would be to tone down the conclusions on this point

The reviewers are correct: additional evidence is necessary to conclude that the retrograde transport is required for Ret-mediated axon outgrowth. However, upon reflection we were uncertain if this particular experiment would address the question of whether retrograde transport is required for Ret-mediated axon growth in our system. Our double mutant data shows no significant difference between different permutations of *jip3* genotypes in *ret* homozygous mutants. Attempting rescue by adding back Ret to the double homozygous mutant would result in a condition essentially the same as looking at the Jip3 mutant individually (where Ret is normally present), and mRNA-based rescue is mosaic and variable.

In the revised manuscript, we elected to address a related question: what is the contribution of Jip3-mediated retrograde transport (versus local Ret activity) to the pLL axon growth in the *ret* mutant phenotype? To achieve this, we acutely blocked Ret signaling in *jip3* mutants using pharmacological approach. First, we quantified axon truncation in *jip3* mutants (new Figure 3I) to show that *jip3* mutants alone have a less severe axon truncation than *ret* mutants alone or *ret/jip3* doublemutants. We have provided additional quantification of *jip3* mutant axon truncation in the manuscript and noted that *ret* mutant axon truncations are generally more severe. This suggests that loss of Jip3 does not account for all Ret-mediated axon growth, possibly because local Ret signaling is responsible for some aspects of axon outgrowth. Put another way, “adding back” Ret into the *ret/jip3* double mutant provides a partial but not complete “rescue” of axon growth.

Next, we tested whether application of a pharmacological Ret inhibitor, sorafenib (which prevents its activating phosphorylation (Plaza-Menacho et al., 2007)), would affect the extent of axon truncation in *jip3* mutants. Ret inhibitor application would prevent activation of Ret in the extending growth cone of *jip3* mutants during axon growth. Sorafenib treatment in wild-type siblings induced a more severe truncation phenotype than that observed in untreated *jip3* mutants. Furthermore, loss of *jip3* did not exacerbate the sorafenib-induced axon truncation phenotype. Combined with our other observations, these data suggest that Jip3, and presumably Jip3-mediated retrograde transport, is required for a significant portion, but not all, of Ret-mediated pLLG axon outgrowth. This additional data is illustrated in Figures 3I, 6C, Figure 6—figure supplement 1. We also added discussion to the manuscript and altered conclusions accordingly (Discussion, third paragraph, as noted in our response to reviewer point #2).

5) The specificity of Myo10Ab staining (Figure 7D'-E") should be confirmed in myo10I1 mutants. In addition, it would be interesting to determine whether Myo10 expression changes in the cell body and axons of Jip3 mutants with blunted RET long-range signaling.

We do not have a *myo10l1* germ line mutants at hand at this point (this strain is 3 to 6 months away). The data in our manuscript take advantage of transient Myo10l1 CRISPR injections. In the revised manuscript, we addressed Ab specificity by mosaic overexpression of tagged Myo10l1-eGFP, followed by immunostaining with our Myo10 antibody. We observe high levels of Myo10 antibody staining in neurons expressing Myo10l1eGFP (Figure 7—figure supplement 1A).

6) The conclusion that retrograde Ret signaling and transcriptional activation of myo10 is based on quantification of fluorescence in situ hybridization intensity. This may not be the best approach because absolute fluorescence intensity can vary between embryos. Could the authors show a difference in mRNA levels with quantitative real-time PCR?

Indeed, quantitative real-time PCR is a standard technique to assay for differences in mRNA expression. However, in our case, whole embryo qRT-PCR is not sensitive enough to address *myo10l1* expression changes due to the small number of cells whose *myo10l1* expression is impacted compared to the entirety of the embryo. We have previously isolated pLL ganglia (~50 cells) and attempted various methods of producing and amplifying a cDNA library, but, unfortunately, the amount of material present was insufficient for qRT-PCR.

In our study, we account for fluctuations in fluorescence by subtracting averaged background staining in each individual embryo to get normalized fluorescence intensity values. We process all embryos in the same tube to ensure identical staining conditions and minimize variation. We then image/score each batch of embryos blind and genotype afterwards. This is described in the Materials and methods section.